# Communication Cost Reduction for Subgraph Counting under Local Differential Privacy via Hash Functions

**Quentin Hillebrand** *quentin-hillebrand@g.ecc.u-tokyo.ac.jp*
*The University of Tokyo*

**Vorapong Suppakitpaisarn** *vorapong@is.s.u-tokyo.ac.jp*
*The University of Tokyo*

**Tetsuo Shibuya** *tshibuya@hgc.jp*
*The University of Tokyo*

**Reviewed on OpenReview:** *https://openreview.net/forum?id=N1J236mepp*

## Abstract

We suggest the use of hash functions to cut down the communication costs when counting triangle and other subgraphs under edge local differential privacy. While various algorithms exist for computing graph statistics — including the count of subgraphs — under the edge local differential privacy, many suffer with high communication costs, making them less efficient for large graphs. Though data compression is a typical approach in differential privacy, its application in local differential privacy requires a form of compression that every node can reproduce. In our study, we introduce linear congruence hashing. Leveraging amplification by sub-sampling, with a sampling size of $s$, our method can cut communication costs by a factor of $s^2$, albeit at the cost of increasing variance in the published graph statistic by a factor of $s$. The experimental results indicate that, when matched for communication costs, our method achieves a reduction in the $\ell_2$-error by up to 1000 times for triangle counts and by up to $10^3$ times for 4-cycles counts compared to the performance of leading algorithms.

## 1 Introduction

*Differential privacy* (Dwork, 2006; Dwork et al., 2014) has emerged as a benchmark for protecting user data. To meet this standard, it is necessary to obfuscate the publication results. This can be achieved by adding small noise (Dwork et al., 2006) or altering the publication outcomes with small probability (McSherry & Talwar, 2007).

In differential privacy, it is typically assumed that there is a complete dataset available. After statistical analyses are done, obfuscation is applied to the results. However, there can be leaks of user information during the data collection or storage phases. To address these concerns, a variant of differential privacy, termed *local differential privacy* (Cormode et al., 2018; Evfimievski et al., 2003), has been introduced. Here, rather than starting with a full dataset, each user is prompted to disguise their data before sharing it. As a result, the received dataset is not perfect. Numerous studies, like those referenced in Li et al. (2020); Asi et al. (2022), have aimed to extract accurate statistical insights from these imperfect datasets. The local differential privacy has been practically employed by several companies to guarantee the privacy of their users' information. Those companies include Apple, Microsoft, and Google (Differential Privacy Team, Apple, 2017; Ding et al., 2017; Erlingsson et al., 2014).

Many studies on local differential privacy focus on tabular datasets, but there is also significant research dedicated to publishing graph statistics (Sajadmanesh & Gatica-Perez, 2021; Ye et al., 2020). For graph-based inputs, such as social networks, the prevalent privacy standard is *edge local differential privacy* (Qin

et al., 2017). In this framework, users are asked to share a disguised version of their adjacency vectors. These vectors are bit sequences which show mutual connections within the network. For instance, in a social network comprising $n$ users, user $v_i$ would provide their adjacency vector, denoted as $a_i = [a_{i,1}, \ldots, a_{i,n}] \in \{0, 1\}^n$. Here, $a_{i,j} = 1$ implies that users $v_i$ and $v_j$ are connected, while $a_{i,j} = 0$ means there is no connection between them.

One widely adopted method for obfuscation is the *randomized response* (Warner, 1965; Mangat, 1994; Wang et al., 2016). In this approach, users are prompted to invert each bit in their adjacency vector, denoted as $a_{i,j}$, based on a specific probability. However, while the method is straightforward, its application to real-world social networks presents challenges. Typically, in practical social networks, an individual may have at most a few thousand friends, implying that the majority of $a_{i,j}$ values are zeroes. When employing the randomized response technique and flipping each bit of $a_{i,j}$ based on the designated probability, a significant number of zero bits get inverted to one. This phenomenon can distort the resulting graph statistics considerably (Mukherjee & Suppakitpaisarn, 2023; Mohamed et al., 2022).

Beyond the publication of full graphs via randomized response, there is a growing interest in subgraph counting queries under local differential privacy. The challenge was first highlighted by Imola et al. (2021), who proposed two key algorithms for $k$-star and triangle counting. Building on $k$-star counting, Hillebrand et al. (2023) later introduced an unbiased and refined algorithm. For triangle counting, several notable contributions include Imola et al. (2022a), which addressed the communication cost, Liu et al. (2022; 2024b), which delved into scenarios where users have access to a 2-hop graph view, and Eden et al. (2023), which studies the lower bound of the additive error. For larger subgraphs, He et al. (2024) gave a local differentially private algorithm to count butterflies on bipartite graphs, Betzer et al. (2024) gave an algorithm to count the number of walks under the privacy notion, Hillebrand et al. (2025) proposed an algorithm to count odd-length cycles on degeneracy-bounded graphs, and Suppakitpaisarn et al. (2025) devised an algorithm to count any graphlet of size $k$. Moving to the shuffle model, Imola et al. (2022b) gave algorithms for triangle and 4-cycle counting. To conclude, Dhulipala et al. (2022) gave an approximation method for identifying the densest subgraph, rooted in locally adjustable algorithms.

Most of the works mentioned in the previous paragraph use the two-step mechanism (Imola et al., 2021; 2022a;b). In this mechanism, users are required to download the adjacency vectors of all other participants. They then compute graph statistics locally using their genuine adjacency vector and the obfuscated vectors of their peers. Although this technique substantially increases the accuracy for various graph statistics, such as the count of triangles, it comes with the drawback of demanding a vast amount of data download. Each user incurs a communication cost of $\Theta(n^2)$. Given that the user count, $n$, can reach several billion in real-world scenarios, this download demand is not affordable for most.

Instead of using adjacency vectors, users might consider exchanging adjacency lists. This approach involves each user transmitting and receiving only a list of adjacent node pairs. If we assume that a user can have a limited, constant number of friends, then the bit-download requirement for the original graph becomes $\Theta(n \log n)$. However, given the surge in edge count to $\Theta(n^2)$ due to the randomized response, the communication cost associated with downloading the obfuscated adjacency list shoots up to $\Theta(n^2 \log n)$. This means that, post-obfuscation, the adjacency list fails to offer any communication efficiency benefits over the adjacency vector.

There are several techniques proposed to reduce the communication cost for the two-step mechanism such as the asymmetric randomized response (Imola et al., 2022a), the degree-preserving randomized response (Hidano & Murakami, 2024), the degree-preserving exponential mechanism (Adhikari et al., 2020), or technique based on the compressive sensing (Li et al., 2011). However, we strongly believe that the communication cost could be further reduced both in theory and practice.

## 1.1 Our Contributions

To mitigate the communication overhead, we suggest employing a *compression* technique on the adjacency vector or list prior to its transmission to the central server.

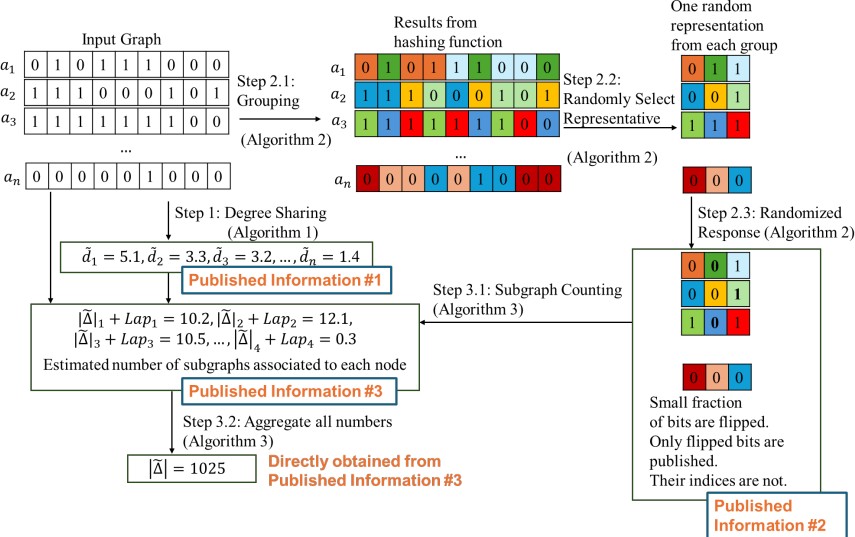

Figure 1: Overview of our mechanism, GroupRR: the information within the green box represents the data published by users. By the amplification-by-sub-sampling theorem, the randomized response in Step 2.3 has a bit-flip probability reduced by approximately a factor of $s$ relative to the standard randomized response. This helps reducing the error we have in our proposed algorithm.

The concept of database compression is not novel within the realm of differential privacy. Numerous mechanisms have been developed to enhance the precision of statistics derived from tabular data through database compression using sampling. One of the most prominent among these is the SmallDB algorithm (Blum et al., 2013). There have also been initiatives that harness the Kronecker graph to refine the accuracy of published graph statistics (Paul et al., 2020).

On the contrary, implementing database compression for the two-step mechanism under edge local differential privacy poses greater challenges. Imagine we compress the adjacency list $[a_{i,1}, \ldots, a_{i,n}]^t$ down to $[a_{i,c_1}, \ldots, a_{i,c_m}]^t$, with $m \ll n$, via sampling, and then disseminate this compressed list to all users. When other users utilize this compressed list for local calculations in the second step, they must be able to replicate the sampling outcome. We therefore need a function which can be replicated with small communication cost. This requirement inspired our proposition to utilize *linear congruence hashing* (Thomson, 1958; Rotenberg, 1960) for the sampling process.

Our proposal is outlined in Figure 1. Using linear congruence hashing, we can evenly partition the node set $\{v_1, \ldots, v_n\}$ into $m$ subsets. Let us denote these subsets as $S_1, \ldots, S_m$, each having a size of $s$. The index $c_i$, corresponding to the only value from the group sent to the server, is derived from $S_i$ with a uniform probability of $1/s$. Since all data is transmitted from the users with a probability of $1/s$, we can leverage the theorem on amplification by sub-sampling (Balle et al., 2018) to diminish the bit-flipping probability in the randomized response mechanism by a factor of $1/s$. Consequently, the count of non-zero entries in the bit vector $[a_{i,c_1}, \ldots, a_{i,c_m}]^t$ is roughly $m/s$, which is approximately equal to $n/s^2$. By employing a sampling size of $s$, the communication overhead can be reduced by a factor of $s^2$.

While our publication's variance might increase by up to a factor of $s$ for certain graph statistics, our experimental results demonstrate that, for a fixed download cost, our algorithm can reduce the $\ell_2$-error in triangle counts by a factor of 1000.

A valid inquiry might be whether we would obtain a comparable outcome by deterministically choosing the same disjoint sets $S_1, \ldots, S_m$ for every user $v_i$. While this approach could simplify the algorithm by eliminating the need for hash functions, our findings indicate that deterministic set selection leads to a significant variance in the estimations during the mechanism's second step. Hence, using the linear congruence hashing is vital for our compression.

The main contribution we present in the article is the first purely local differential private mechanism for graph statistics that leverages amplification by sub-sampling. Additionally, we demonstrate the generality of the mechanism and its efficiency as it performs better than state of the art by several orders of magnitudes for several triangle and other subgraph counting tasks.

### 1.2 Related Works

Hashing functions have been utilized in the domain of differential privacy by Wang et al. (2017) to develop binary local hashing (BLH) and optimal local hashing (OLH) for frequency estimation under local differential privacy. However, the work focuses on tabular information, while our work focuses on graph information. The proposed algorithm is totally different because of the difference in data structure and privacy definition. Additionally, while the previous method use hash functions to compress user data, they do not support amplification by sub-sampling, which is a feature of our proposed method. Thus, our compression rate is larger than the previous work.

A recent study on communication reduction in local differential privacy (Liu et al., 2024a) successfully simulated any randomizer for local differential privacy using only $\mathcal{O}(\varepsilon)$ communication bits. However, similar to BLH and OLH methods, their approach depends on the characteristics of tabular data and traditional local differential privacy, making it inapplicable to our scenario where edge-local differential privacy is employed.

## 2 Preliminaries

### 2.1 Hash functions

Hash functions (Knuth, 1997) map keys from a specified key space to a designated hash space. In this paper, unless mentioned differently, the key space is represented by $[|0, n-1|]$ and the hash space by $[|0, m-1|]$.

We consider a hashing scheme called linear congruence hashing (Thomson, 1958; Rotenberg, 1960) in this paper. The scheme can be defined as $\mathcal{H} = \{h_{\mathsf{a},\mathsf{b}} | \mathsf{a} \in [|1, \mathsf{p}-1|], \mathsf{b} \in [|0, \mathsf{p}-1|]\}$ when $\mathsf{p}$ is a prime number greater than $n$ and $h_{\mathsf{a},\mathsf{b}}(\mathsf{x}) = ((\mathsf{a}\mathsf{x} + \mathsf{b}) \bmod \mathsf{p}) \bmod m$. Notice that, for all $\mathsf{a}, \mathsf{b}$ and $\mathsf{k} \in [|0, m-1|]$, we have $\lfloor \frac{n}{m} \rfloor \leq |\{\mathsf{x} : h_{\mathsf{a},\mathsf{b}}(\mathsf{x}) = k\}| \leq \lceil \frac{n}{m} \rceil$. In other words, the number of key in a given bin is either $\lfloor \frac{n}{m} \rfloor$ or $\lceil \frac{n}{m} \rceil$.

### 2.2 Edge Local Differential Privacy

Let the set of users be $\{v_1, \ldots, v_n\}$. Each user $v_i$ has own adjacency list $a_i = [a_{i,1}, \ldots, a_{i,n}]^t \in \{0,1\}^n$ where $a_{i,j} = 1$ if $\{v_i, v_j\} \in E$ and $a_{i,j} = 0$ otherwise. For any $a_i, a'_i \in \{0,1\}^n$, let $d(a_i, a'_i)$ be the $\ell_1$-distance between the two vectors. To define the privacy notion, we first give the following definition:

**Definition 1** (Local differentially private query). Let $\epsilon > 0$. A randomized query $\mathcal{R}$ is said to be $\epsilon$-edge locally differentially private for node $i$ if, for any pair of adjacency lists $a_i$ and $a'_i$ where $|a_i - a'_i| \leq 1$, and for any possible set of outcomes $S$, we have that

$$\Pr[\mathcal{R}(a_i) \in S] \leq e^\epsilon \Pr[\mathcal{R}(a'_i) \in S].$$

Then, the definition of the edge local differential privacy is as follows:

**Definition 2** (Edge local differential privacy Qin et al. (2017)). An algorithm $\mathcal{A}$ is defined as $\epsilon$-edge locally differentially private if, for any user $i$ and for any possible set of queries $\mathcal{R}_1, \ldots, \mathcal{R}_\kappa$ which $\mathcal{A}$ posed to user $i$, where each query $\mathcal{R}_j$ is $\epsilon_j$-edge locally differentially private (for $1 \leq j \leq \kappa$), the condition $\epsilon_1 + \cdots + \epsilon_\kappa \leq \epsilon$ is satisfied.

In this setting, $\varepsilon$ represents the degree of privacy protection and is termed the privacy budget. A lower privacy budget value indicates a stricter privacy limitation on published data. Each user possesses an adjacency list denoted by $a_i$. Before transmitting the result to the central server, they apply the randomized function $\mathcal{R}$. Subsequently, the central server aggregates this data using the aggregator $\mathcal{A}$. According to Definition 2, if a mechanism $\mathcal{M}$ is $\varepsilon$-edge differentially private, significant information of $a_i$ cannot be obtained from $\mathcal{R}(a_i)$ once it is transferred and stored on the central server.

One of the most important properties of the edge local differential privacy is the composition theorem.

**Theorem 1** (Composition Theorem (Dwork et al., 2010)). *Let $\mathcal{M}_1, \ldots, \mathcal{M}_p$ be edge local differentially private mechanism with privacy budget $\varepsilon_1, \ldots, \varepsilon_p$. Then, the mechanism $\mathcal{M}_p \circ \cdots \circ \mathcal{M}_1$ is $(\varepsilon_1 + \cdots + \varepsilon_p)$-edge local differentially private.*

### 2.3 Basic Mechanisms

Numerous mechanisms have been suggested to meet Definition 2 (Li et al., 2022; Hou et al., 2023). Among them, the edge local Laplacian mechanism (Hillebrand et al., 2023) is an approach to offer privacy when we request each user to give real numbers to the aggregator. The Laplacian mechanism can be defined as in the following definition. It can be shown that the mechanism is $\varepsilon$-edge local differential private.

**Definition 3** (Edge Local Laplacian Mechanism (Hillebrand et al., 2023)). For $f : \{0,1\}^n \to \mathbb{R}$, the global sensitivity of $f$ is $\Delta_f = \max\limits_{d(a,a')=1} \|f(a) - f(a')\|_1$. For $\varepsilon > 0$, let $X$ be a random variable drawn from $\text{Lap}(\varepsilon/\Delta_f)$. The edge local Laplacian mechanism for $f$ with privacy budget $\varepsilon$ is the mechanism for which the randomized function $\mathcal{R}$ is defined as $f(a) + X$.

When we request users to send a vector of bits (such as the whole adjacency vector) to the aggregator, the edge local Laplacian mechanism is not the most suitable. Instead, a frequently used method is the randomized response (Warner, 1965; Mangat, 1994; Wang et al., 2016), as detailed in the following definition.

**Definition 4** (Randomized Response (Warner, 1965; Mangat, 1994; Wang et al., 2016)). For $\varepsilon > 0$, the randomized response with privacy budget $\varepsilon$ is a mechanism which the randomized function $\mathcal{R} : \{0,1\}^n \to \{0,1\}^n$ is a function such that $\mathcal{R}(a_{i,1}, \ldots, a_{i,n}) = (\text{RR}(a_{i,1}), \ldots, \text{RR}(a_{i,n}))$ where

$$\mathbb{P}\left(\text{RR}(a_{i,j}) = 1\right) = \begin{cases} \frac{e^\varepsilon}{1+e^\varepsilon} & \text{if } a_{i,j} = 1 \\ \frac{1}{1+e^\varepsilon} & \text{if } a_{i,j} = 0. \end{cases} \tag{1}$$

### 2.4 Smooth Sensitivity (Nissim et al., 2007)

The traditional Laplacian mechanism adjusts noise based on the global sensitivity, which is determined from the worst possible outcome for a selected function. It does not consider the specific adjacency list. Therefore, in certain situations, the global sensitivity might be much greater than the function's changes due to slight update in the real adjacency list. Noticing this, Nissim et al. (2007) created a mechanism where the noise relates to the smooth sensitivity, which varies depending on the instance. Before we define smooth sensitivity, we first need to explain local sensitivity and sensitivity at a distance $k$.

**Definition 5** (Local Sensitivity). For $f : \{0,1\}^n \to \mathbb{R}$, the local sensitivity of $f$ on $a \in \{0,1\}^n$ is $\text{LS}_f(a) = \max\limits_{a':d(a,a')=1} \|f(a) - f(a')\|_1$.

The local sensitivity of $f$ on $\{0,1\}^n$ measures the largest change in $f$ due to a single edge alteration when starting with the adjacency list $a$. The sensitivity of $f$ at distance $k$ expands on this idea by looking at the changes resulting from a single alteration in the adjacency lists that are within $k$ modifications from $a$.

**Definition 6** (Sensitivity at Distance $k$). For $f : \{0,1\}^n \to \mathbb{R}$, the sensitivity of $f$ at distance $k$ at $a$ is $A^{(k)}(a) = \max\limits_{a':d(a,a')\leq k} \text{LS}_f(a')$.

We are now ready to define the smooth sensitivity.

**Definition 7** (Smooth Sensitivity). For $f$ a function and $\beta > 0$, the $\beta$-smooth sensitivity of $f$ at $a$ is $S^*_{f,\beta}(a) = \max\limits_k e^{-\beta k} A^{(k)}(a)$

We are now ready to define the mechanism based on the smooth sensitivity. It is shown that the mechanism is $\varepsilon$-edge local differential private.

**Definition 8** (Smooth Sensitivity Mechanism). Let $\gamma > 1$ and $h(z)$ be a probability distribution proportional to $1/(1 + |z|^\gamma)$. We will denote $Z_\gamma$ the random variable drawn from $h$. For $\varepsilon > 0$ and $\gamma > 1$, the smooth

sensitivity mechanism for $f$, denoted by $\mathcal{M}$, is a mechanism with the randomized function $\mathcal{R}(a) = f(a) + \frac{4\gamma}{\varepsilon} S^*_{f,\varepsilon/\gamma}(a) \cdot Z_\gamma$.

This result was later improved in Yamamoto & Shibuya (2023) who proved that $\mathcal{R}(a)$ could be changed to $f(a) + \frac{2(\gamma-1)}{\varepsilon} S^*_{f,\varepsilon/2(\gamma-1)}(a) \cdot Z_\gamma$.

The mechanism operates effectively for all values of $\gamma > 1$. Nonetheless, it is important to highlight that when $\gamma \leq 3$, we do not know a way to calculate the variance of the random variable $Z_\gamma$. Consequently, we will adopt $\gamma = 4$ for this article, where the variance of $Z_\gamma$ is one.

## 3 Previous Mechanism

Let us suppose the the original graph is $G = (V, E)$ when $V = \{v_1, \ldots, v_n\}$ is a set of nodes or users and $E \subseteq \{\{u, v\} : u, v \in V\}$ is a set of edges or relationships. We can publish $G$ using the randomized response mechanism defined in the previous section. Given that $a'_{i,j} = \texttt{RR}(a_{i,j})$ obtained from the randomized response, we can define a graph $G' = (V, E')$ with its adjacency matrix given by $A' = (a'_{i,j})_{1 \leq i,j \leq n}$. One can determine the number of subgraphs in $G'$ by counting them; however, this method introduces a significant error (Imola et al., 2022a). Specifically, the error magnitude is $\mathcal{O}\left(\frac{C_4(G)}{\varepsilon^2} + \frac{n^3}{\varepsilon^6}\right)$, which is large in relation to $n$ and becomes prohibitive as $\varepsilon$ decreases, particularly when aiming for higher levels of privacy.

### 3.1 Two-Step Mechanism

To mitigate the error, Imola et al. (2021) introduced a two-step mechanism to release the count of triangles. A tuple of three nodes, represented as $(v_i, v_j, v_k)$ where $i > j > k$ is a triangle if $\{v_i, v_j\}, \{v_i, v_k\}, \{v_j, v_k\} \in E$. We denote the set of triangles in $G$ by $\Delta_G$. Furthermore, for tuples in $\Delta_G$ where the initial element is $v_i$, we denote them as $\Delta_i$. We have that $|\Delta_G| = \sum_i |\Delta_i|$.

The first step, represented by $\mathcal{M}_1$, involves releasing the obscured graph $G' = (V, E')$ produced by the randomized response. In the second step, represented by $\mathcal{M}_2$, every user $v_i$ provides an estimated value for $|\Delta_i|$ that is based on both the real adjacency vector $a_i$ and the distorted graph $G'$. Let $\Delta'_i = \{(v_j, v_k) : \{v_i, v_j\}, \{v_i, v_k\} \in E, \{v_j, v_k\} \in E', \text{ and } i > j > k\}$. Consider a function $f$ defined as $f(a_i) = |\Delta'_i|$. The randomized funtion of $\mathcal{M}_2$, represented as $\mathcal{R}_2$, aligns with what is used in the edge local Laplacian mechanism. Specifically, users are asked to add a Laplacian noise into $f(a_i)$. The mechanism's aggregator then sums the outcomes derived from these randomized functions.

The two-step mechanism can significantly increase the precision of the triangle counting. However, it has a large variance. Consider $i > i' > j > k$ such that $\{v_i, v_j\}, \{v_i, v_k\}, \{v_{i'}, v_j\}, \{v_{i'}, v_k\} \in E$ and $\{v_j, v_k\} \notin E$. If, by the randomized response, $\{v_j, v_k\} \in E'$, that would contribute two to the number of triangles. When the input graph has a large number of $C_4$, the variance becomes large. The authors of Imola et al. (2022a) propose a technique called four-cycle trick to reduce the variance. Define $\Delta''_i := \{(v_j, v_k) \in \Delta'_i : \{v_i, v_k\} \in G'\}$. Instead of having $f(a_i) = |\Delta'_i|$, they use $f(a_i) = |\Delta''_i|$. For the rest of the article, we will use ARROne to denote the version of ARR that uses this 4-cycle trick.

### 3.2 Communication Cost

In many real-world social networks, each user is typically connected to a constant number of other users (Barabási, 2013). This means the number of connections (or 'ones' in the adjacency list $a_i$) is a lot fewer than $n$. So, instead of sending their entire adjacency list $a_i$ to the aggregator, user $v_i$ can just share the list of those directly connected to them. This approach reduces the communication cost from $n$ bits down to $\Theta(\log n)$.

However, when using randomized response mechanisms, every bit in $a_i$ has a constant probability of being flipped. The resulting adjacency list $a'_i$ will have roughly $\Theta(n)$ ones. So, the communication-saving technique mentioned earlier will not work here.

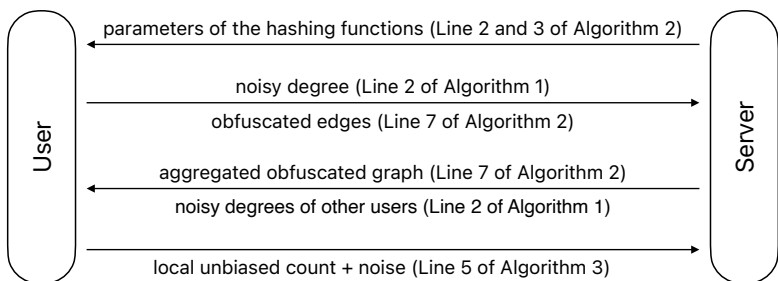

Figure 2: Details of the interactions between each user and the server during the execution of our mechanism

The situation gets more difficult with the two-step mechanism discussed earlier. In this approach, each user needs to access the entire distorted graph $A'$. The user $v_i$ cannot choose to access only the edge $\{v_j, v_k\}$ such that $\{v_i, v_j\}, \{v_i, v_k\} \in E$ as that will reveal the existence of those two edges. This means they have to download the whole distorted graph, which requires about $\Theta(n^2)$ bits. Such a cost becomes unrealistic for social networks with millions of nodes.

The four-cycle trick mentioned in the previous section can help reducing the communication cost. As a triangle $(v_j, v_k) \in \Delta''_i$ only if $\{v_i, v_k\} \in E'$, user $v_i$ can calculate $|\Delta''_i|$ without accessing the information about whether $\{v_j, v_k\} \in E'$ or not when $\{v_i, v_k\} \notin E'$. Still, we think that the savings from the trick are not substantial enough to make the two-step method viable for big graphs.

## 4 Our Mechanism: Group Randomized Response

In this section, we introduce a mechanism designed for publishing graphs in a distributed environment under the edge local differential privacy. We have termed this approach the *group randomized response* (GroupRR). The mechanism is a three-step process:

1. **Degree Sharing Step**: In this step, each user shares their degree protected by the local Laplacian mechanism.

2. **Group Randomized Response Step**: In this step, a sampled adjacency list is published by each user. First, nodes and edges are categorized into groups using the linear congruence hashing function. Then, one value is sampled from each group and only this value is obfuscated and sent to the central server.

3. **Counting Step:** Subgraphs are counted based on the information from the representatives. Then, each user shares the number of subgraphs protected by the local Laplacian mechanism.

Figure 1 provides an overview of the various steps of the mechanism, while Figure 2 emphasizes the communication between users and the central server.

Thanks to the sampling created by the formation of groups, this mechanism achieves a notably reduced communication cost. Moreover, the privacy budget amplification from sampling (Balle et al., 2018) further decreases the probabilities of bit-flipping during the randomized response phase, further reducing the communication cost.

### 4.1 Degree Sharing Step

During this step, each user $v_i$ calculates the count of their connections, specifically $d_i = |\{j < i : \{i, j\} \in E\}|$. We call $d_i$ as *low degree* in this paper. They then share the low degree using the local Laplacian mechanism

with the privacy budget of $\varepsilon_0$. The shared count, denoted by $\tilde{d}_i$, will be used only for adjusting bias in the fourth step. If there are a lot of users ($n$ is large), any errors from this count are minor even if a large Laplacian noise is added. Thus, we can choose a relatively small value for $\varepsilon_0$. This step is described as in Algorithm 1.

---

**1** | **Function** DegreeSharing
  |     **Input:** Graph $G = (V, E)$, privacy budget $\varepsilon_0$
  |     **Output:** Estimated degree $\tilde{d}_i$ of each user
**2** |     [**User** $i$] Calculate and send $\tilde{d}_i \leftarrow d_i + \text{Lap}(\frac{1}{\varepsilon_0})$ to the central server

**Algorithm 1:** The first step of our proposed method, degree sharing step

---

### 4.2 Group Randomized Response Step

The goal of this step is to publish an obfuscated adjacency list in a communication efficient way. Our solution is to form random groups of edges that are of similar sizes. It is crucial that this randomization can be reproduced, and that the division is easily shareable among all participants. To achieve such a grouping that meets these criteria, we employ the linear congruence hashing, which is discussed in Section 2. Once that grouping is done, one value is sampled per group. This value is then obfuscated with the randomized response mechanism and broadcasted. This step is described in Algorithm 2.

At Line 2 of the algorithm, the central server calculates $p \in \mathbb{N}$, which represents the smallest prime number greater than $n$, the graph's size. Then, at Line 3, for each node $v_i$ when $i \in [[1, n]]$, the server randomly generates two values: $\theta_i$ from the range $[[1, p-1]]$ and $\phi_i$ from the range $[[0, p-1]]$. These values act as coefficients for the linear congruence hashing. After generating these coefficients, the server shares them, along with $p$, with all users.

Recall the definition of $h_{\mathsf{a,b}}$ in Section 2. Let $s \in \mathbb{N}^*$ be a parameter called *sampling size* which will be the size of each group, $m$ be an integer such that $m = \lceil p/s \rceil$, and let $h_i(j) = h_{\theta_i, \phi_i}(j) = ((\theta_i \cdot j + \phi_i) \bmod p) \bmod m$. At Line 4, each user $v_i$ classifies the set of users $\{v_1, \ldots, v_n\}$ into $m$ disjoint groups, denoted by $S_{i,1}, \ldots, S_{i,m}$. The group $S_{i,t}$ can be defined as $\{v_j : h_i(j) = t\}$. By the property of the linear congruence hashing, we have that $s - 1 \leq |S_{i,t}| \leq s$. To make sure that all the sets have size $s$, we add a dummy element to all $S_{i,t}$ with size $s - 1$. There is no edge between $v_i$ and the dummy node.

At Line 5, we then choose a representative for each group, $S_{i,t}$. Every member of the group has an equal probability of being selected, which is $1/s$. Let us call the chosen member from group $S_{i,t}$ as $v_{c_t}$. When $c_t$ is chosen from the group containing $k$, we will employ the notation $a_{i,h_i(k)} := a_{i,c_t}$ for convenience. Only the value of the representative edge $a_{i,c_t}$ is published via the randomized response mechanism, and the actual selected index $c_t$ is not shared. We introduce this quantity only to simplify the description and analysis of the algorithm.

It should be emphasized that even though the hashing coefficients are determined by the central server, this does not pose a privacy risk. This is because the hashing functions are employed to form groups that are intentionally public. The privacy-sensitive element is the representative value chosen for each group, and the selection of these representatives does not involve the hash functions.

Each user then sends the values $a_{i,c_1}, \ldots, a_{i,c_m}$ to the aggregator using the randomized response mechanism with a privacy budget of $\varepsilon' = \ln(1 + s(e^{\varepsilon_1} - 1))$. It is noteworthy that this privacy budget, $\varepsilon'$, is considerably greater than the $\varepsilon_1$ budget used in the traditional method. This results in a much smaller probability of flipping bits in the adjacency vector.

The obfuscated values sent from $v_i$ to the central system are denoted as $a'_{i,c_1}, \ldots, a'_{i,c_m} \in \{0, 1\}$. The aggregator then releases these values, making them available for each user in the mechanism's final step. In the earlier approach, users had to know the list of $v_j$ such that $a'_{i,j} = 1$. Now, they only need to know the list of $t$ such that $a'_{i,c_t} = 1$. This significant reduces the number of bits users have to download from the central server. We will conduct the analysis on the number of bits which each user needs to download in the next section.

To ensure that each edge is published only once, each user only publishes connections with users who have a smaller index than their own. To enforce this condition, in Line 6, we assign $a'_{i,c_t}$ to zero when $c_t$ is greater than $i$.

---

**1** **Function** GroupRandomizedResponse
   **Input:** Graph $G = (V, E)$, sampling size $s$, privacy budget $\varepsilon_1$
   **Output:** The grouped and obfuscated adjacency list $(a'_{i,c_1}, \ldots, a'_{i,c_m})$ for each user
**2**    [**Server**] Calculate and broadcast $p$ the smallest prime number larger than $n$
**3**    [**Server**] For all $i \in [1, n]$, randomly select and broadcast $(\theta_i, \phi_i)$ from $[1, p-1] \times [0, p-1]$
**4**    [**User** $i$] Recall the definition of $h_{a,b}$ in Section 2. Let $m$ be an integer such that $m = \lceil p/s \rceil$, and let $h_i(j) = h_{\theta_i, \phi_i}(j) = ((\theta_i \cdot j + \phi_i) \bmod p) \bmod m$. For each $t \in [1, m]$, $S_{i,t} \leftarrow \{v_j \mid h_i(j) = t\}$;
**5**    [**User** $i$] Choose randomly $v_{c_t}$ from the set $S_{i,t}$
**6**    [**User** $i$] For all $t \in [1, m]$, if $c_t > i$, $a_{i,c_t} \leftarrow 0$;
**7**    [**User** $i$] Perform the randomized response with privacy budget $\varepsilon' = \ln(1 + s(e^{\varepsilon_1} - 1))$ on $(a_{i,c_1}, \ldots, a_{i,c_m})$. The result $(a'_{i,c_1}, \ldots, a'_{i,c_m})$ is then broadcasted to all users (We note that only $a'_{i,c_i}$ for all $1 \le i \le m$ is published. The index $c_i$ is not.);

**Algorithm 2:** The second step of our proposed method, group randomized response step

---

### 4.3 Counting Step

The final step of our method is outlined in Algorithm 3. In Lines 2-3, we update the value of $a'_{i,c_t}$ to $\tilde{a}_{i,c_t} = \omega_i \cdot a'_{i,c_t} - \tilde{\sigma}_i$. In the next section, we will show that this adjustment effectively removes the bias in our subgraph counting. We then use the value of $\tilde{a}_{i,c_t}$ to estimate the number of subgraphs associated with each node. The calculation method varies depending on the specific subgraph. In Line 4 of the algorithm, we detail the process for triangle counting. However, we can modify this line to calculate other graph statistics. Following this, in Line 5, we apply either the local Laplacian mechanism or the smooth sensitivity mechanism to obfuscate the result from Line 4 before sending it to the central server. At Line 6, the central server aggregates the results received from all users and publishes the summation as the final counting result.

It is crucial to understand that we do not assume the degree $d_i$ is public. Rather, we utilize an estimation of $d_i$, referred to as $\tilde{d}_i$, which is disclosed in the initial step of our mechanism, for the calculation of $\tilde{\sigma}_i$ at Line 2 of the algorithm.

---

**1** **Function** Counting
   **Input:** Graph $G = (V, E)$, grouped and obfuscated adjacency list $(a'_{i,c_1}, \ldots, a'_{i,c_m})$, privacy budget $\varepsilon_2$, estimated degree $\tilde{d}_i$ of each user.
   **Output:** The estimated number of triangles in the graph
**2**    [**User** $i$] Calculate $\tilde{\sigma}_i = \frac{s-1}{ms-s}\tilde{d}_i + \frac{1}{e^{\varepsilon'}-1}\frac{ms-1}{m-1}$, $\omega_i = \frac{e^{\varepsilon'}+1}{e^{\varepsilon'}-1} \cdot \frac{ms-1}{m-1}$;
**3**    [**User** $i$] For $1 \le t \le m$, calculate $\tilde{a}_{i,c_t} = \omega_i \cdot a'_{i,c_t} - \tilde{\sigma}_i$ ;
**4**    [**User** $i$] Let $W_i := \{(j, k) : j < k < i \text{ and } \{v_i, v_j\}, \{v_i, v_k\} \in E\}$. Calculate $|\tilde{\Delta}_i| = \sum\limits_{(j,k) \in W_i} \tilde{a}_{k,h_k(j)}$;
**5**    [**User** $i$] Obfuscate the value of $|\tilde{\Delta}_i|$ using the local Laclacian mechanism or the smooth sensitivity mechanism with privacy budget $\varepsilon_2$. Then, submit the result to the central server;
**6**    [**Server**] Publish the number of triangle as $\sum_i |\tilde{\Delta}_i|$

**Algorithm 3:** The third step of our method, counting step, when it is used for publishing the number of triangles

## 5 Theoretical Analysis of Our Mechanism

### 5.1 Privacy

We begin by analyzing the privacy guarantees of the group randomized response step (Step 2) in the following lemma. This analysis leverages the amplification by subsampling technique (Balle et al., 2018), defined as follows:

**Lemma 1** (Amplification by Sub-Sampling under Edge Differential Privacy (Balle et al., 2018)). *Let $s > 0$, and define $m = \lceil n/s \rceil$. Consider a randomized sampling function $\mathcal{S} : \{0,1\}^n \to \{0,1\}^m$ such that, given an input $(a_1, \ldots, a_n) \in \{0,1\}^n$, the output is $\mathcal{S}(a_1, \ldots, a_n) = (a'_1, \ldots, a'_m)$, where each $a'_i$ is determined as follows:*

- *For each $1 \leq i < m$, the output $a'_i$ is sampled uniformly at random from the $i$-th block of size $s$, i.e., from indices $\{(i-1)s+1, \ldots, is\}$. That is, for each $j \in \{(i-1)s+1, \ldots, is\}$, we have $a'_i = a_j$ with probability $1/s$.*

- *For $i = m$, let the final block be $\{(m-1)s+1, \ldots, n\}$, which may contain fewer than $s$ elements. Then, for each $j \in \{(m-1)s+1, \ldots, n\}$, we have $a'_m = a_j$ with probability $1/s$, and $a'_m = 0$ with probability $\frac{ms-n}{s}$.*

*Let $\mathcal{M}$ be a randomized algorithm satisfying the following privacy guarantee: for all $a, \hat{a} \in \{0,1\}^n$ such that $|a - \hat{a}| \leq 1$, and for all set $S$,*

$$e^{-\varepsilon'} \leq \frac{\Pr[\mathcal{M}(a) \in S]}{\Pr[\mathcal{M}(\hat{a}) \in S]} \leq e^{\varepsilon'}. \tag{2}$$

*Then, for the composed mechanism $\mathcal{M} \circ \mathcal{S}$, it holds that for all $a, \hat{a} \in \{0,1\}^n$ such that $|a - \hat{a}| \leq 1$, and for all set $S$,*

$$e^{-\varepsilon_s} \leq \frac{\Pr[\mathcal{M}(\mathcal{S}(a)) \in S]}{\Pr[\mathcal{M}(\mathcal{S}(\hat{a})) \in S]} \leq e^{\varepsilon_s}, \tag{3}$$

*where $\varepsilon_s$ is given by $\varepsilon_s = \ln\left(1 + \frac{e^{\varepsilon'}-1}{s}\right)$.*

The amplification by sub-sampling technique was originally proposed in the context of central differential privacy, where each $a_i$ represents sensitive information of user $i$. It has not been previously applied to either local differential privacy or graph statistics. The assumption in (2) requires that $\mathcal{M}$ satisfies $\varepsilon'$-differential privacy, and the conclusion in (3) shows that the composed mechanism achieves $\varepsilon_s$-differential privacy due to the sampling. To the best of our knowledge, we are the first to apply amplification by sub-sampling in the setting of local differential privacy and graph statistics. We consider this application to be one of the main contributions of our work.

The privacy guarantee of the group randomized response step (Step 2) of our algorithm follows from Lemma 1.

**Lemma 2.** *The privacy budget of the group randomized response step is $\varepsilon_1$.*

*Proof.* We begin by observing that the computation of $[a_{i,c_1}, \ldots, a_{i,c_m}]$ in Lines 4–6 of Algorithm 2 can be viewed as a sampling procedure $\mathcal{S}$, as described in Lemma 1, followed by a reordering step, which does not affect the privacy guarantees of the mechanism. The randomized response mechanism applied in Line 7 satisfies the privacy bound given in (2). Consequently, the overall procedure in Algorithm 2, which composes sampling with randomized response, satisfies equation (3), thereby ensuring $\varepsilon_s$-edge local differential privacy as defined in Definition 5. Substituting $\varepsilon' = \ln(1 + s(e^{\varepsilon_1} - 1))$ into the expression for $\varepsilon_s = \ln(1 + \frac{e^{\varepsilon'}-1}{s})$, we conclude that $\varepsilon_s = \varepsilon_1$ and the privacy budget of the group randomized response step is $\varepsilon_1$. $\qquad\square$

We provide the following example to illustrate how this amplification is achieved.

**Example 1.** *Consider the publication of node $\nu_0$'s adjacency list using the group randomized response. We will analyze two cases: (1) $\nu_0$ is connected to none of the $n-1$ other nodes, and (2) $\nu_0$ is connected only to*

$\nu_1$. *Additionally, since the groups created by the hashing functions are public information and independent of the private adjacency list, we assume that $\nu_1$ is in $S_{0,t}$ for both cases.*

*Let $p_1$ and $p_2$ represent the probabilities that $a'_{i,c_t} = 1$ in scenarios (1) and (2), respectively, when using the randomized response mechanism with a privacy budget $\varepsilon'$. In case (1), $a_{i,c_t}$ is never equal to 1, as no nodes are connected to $\nu_0$. By the property of randomized response, $p_1 = \frac{1}{1+e^{\varepsilon'}}$. In case (2), $a_{i,c_t}$ is equal to 1 only when $\nu_1$ is selected as the representative of group $S_{0,t}$ (note that this information is not disclosed). Thus, $a_{i,c_t} = 1$ with probability $\frac{1}{s}$. We have $p_2 = \frac{s-1}{s} \cdot \frac{1}{1+e^{\varepsilon'}} + \frac{1}{s} \cdot \frac{e^{\varepsilon'}}{1+e^{\varepsilon'}}$. We then derive that $\frac{p_2}{p_1} = 1 + \frac{1}{s} \cdot \left( e^{\varepsilon'} - 1 \right)$. By setting $\varepsilon_1$ such that $\varepsilon' = \ln(1 + s(e^{\varepsilon_1} - 1))$, we obtain $\frac{p_2}{p_1} = e^{\varepsilon_1}$. This shows that we achieve $\varepsilon_1$-differential privacy, with $\varepsilon_1 \approx \varepsilon'/s$, demonstrating the privacy budget amplification.*

The final result of our privacy analysis is shown in the following theorem:

**Theorem 2.** *The privacy budget of our mechanism is $\varepsilon_0 + \varepsilon_1 + \varepsilon_2$.*

*Proof.* We directly obtain this result from Lemma 2 and the composition theorem (Theorem 1). □

## 5.2 Expected Value of $\tilde{a}_{i,c_t}$

In the subsequent theorem, we discuss the expected value of $\tilde{a}_{i,c_t}$. The theorem reveals that when an estimate from the counting phase is a linear composition of $\tilde{a}_{i,c_t}$, it remains unbiased. The triangle count estimation in Algorithm 3 is an example of these estimations.

In the following, we will restrict our analysis to the values of $\tilde{a}_{j,h_j(k)}$ where $j > k$. This does not pose any limitation, as $a_{j,k}$ can always be estimated using either $\tilde{a}_{j,h_j(k)}$ or $\tilde{a}_{k,h_k(j)}$. Therefore, for the remainder of this section, we will assume that $j > k$.

Recall that $d_i$ denotes the number of connections between user $v_i$ and nodes with smaller indices, which we refer to as the *low degree* of $v_i$.

Define

$$p_{\mathrm{p},i} := \Pr[a_{j,h_j(k)} = 1 \mid a_{j,k} = 1],$$

which evaluates to

$$p_{\mathrm{p},i} = \frac{s-1}{s} \cdot \frac{d_i - 1}{ms - 1} + \frac{1}{s}.$$

Let

$$p'_{\mathrm{p},i} := \Pr[a'_{j,h_j(k)} = 1 \mid a_{j,k} = 1],$$

which is given by the randomized response mechanism as

$$p'_{\mathrm{p},i} = \frac{e^{\varepsilon'}}{1+e^{\varepsilon'}} \cdot p_{\mathrm{p},i} + \frac{1 - p_{\mathrm{p},i}}{1+e^{\varepsilon'}}.$$

Similarly, define

$$p_{\mathrm{a},i} := \Pr[a_{j,h_j(k)} = 1 \mid a_{j,k} = 0] = \frac{s-1}{s} \cdot \frac{d_i}{ms - 1},$$

and let

$$p'_{\mathrm{a},i} := \Pr[a'_{j,h_j(k)} = 1 \mid a_{j,k} = 0],$$

which satisfies

$$p'_{\mathrm{a},i} = \frac{e^{\varepsilon'}}{1+e^{\varepsilon'}} \cdot p_{\mathrm{a},i} + \frac{1 - p_{\mathrm{a},i}}{1+e^{\varepsilon'}}.$$

In the next lemma, we show a property of $\omega_i$ and $\tilde{\sigma}_i$ calculated in Algorithm 3. From now, denote $\mathbb{E}[\tilde{\sigma}_i]$ by $\sigma_i$.

**Lemma 3.** *For all $i$, $\omega_i = \frac{1}{p'_{\mathrm{p},i} - p'_{\mathrm{a},i}}$ and $\sigma_i = \frac{p'_{\mathrm{a},i}}{p'_{\mathrm{p},i} - p'_{\mathrm{a},i}}$.*

*Proof.* We have

$$\frac{1}{p'_{\mathrm{p},i} - p'_{\mathrm{a},i}} = \frac{1}{\left(\frac{e^{\varepsilon'}-1}{e^{\varepsilon'}+1}\right)(p_{\mathrm{p},i} - p_{\mathrm{a},i})} = \frac{1}{\left(\frac{e^{\varepsilon'}-1}{e^{\varepsilon'}+1}\right)\left(\frac{1}{s} - \frac{s-1}{s} \cdot \frac{1}{ms-1}\right)}.$$

Simplifying the expression gives

$$\frac{1}{p'_{\mathrm{p},i} - p'_{\mathrm{a},i}} = \frac{e^{\varepsilon'}+1}{e^{\varepsilon'}-1} \cdot \frac{ms-1}{m-1} =: \omega_i.$$

On the other hand, we obtain

$$\frac{p'_{\mathrm{a},i}}{p'_{\mathrm{p},i} - p'_{\mathrm{a},i}} = \left(\frac{s-1}{ms-s} \cdot d_i\right) + \left(\frac{1}{e^{\varepsilon'}-1} \cdot \frac{ms-1}{m-1}\right) = \left(\frac{s-1}{ms-s} \cdot \mathbb{E}[\tilde{d}_i]\right) + \left(\frac{1}{e^{\varepsilon'}-1} \cdot \frac{ms-1}{m-1}\right) =: \sigma_i.$$

$\square$

We are now ready the proof the main theorem of this subsection.

**Theorem 3.** *For all $j,k \in \{1,\ldots,n\}$, the expected value of $\tilde{a}_{j,h_j(k)}$ equals $a_{j,k}$.*

*Proof.* When $a_{j,k} = 0$, the expected value of $\tilde{a}_{j,h_j(k)}$ is given by

$$\mathbb{E}\left[\tilde{a}_{j,h_j(k)} \mid a_{j,k} = 0\right] = (\omega_j - \sigma_j) \cdot p'_{\mathrm{a},j} - \sigma_j \cdot (1 - p'_{\mathrm{a},j}) = \omega_j \cdot p'_{\mathrm{a},j} - \sigma_j = 0.$$

Similarly, when $a_{j,k} = 1$, we have

$$\mathbb{E}\left[\tilde{a}_{j,h_j(k)} \mid a_{j,k} = 1\right] = (\omega_j - \sigma_j) \cdot p'_{\mathrm{p},j} - \sigma_j \cdot (1 - p'_{\mathrm{p},j}) = \omega_j \cdot p'_{\mathrm{p},j} - \sigma_j = 1.$$

$\square$

### 5.3 Variance of $\tilde{a}_{i,c_t}$

The previous section demonstrated that our mechanism gives no bias for certain specific publications. In this subsection, we show that the variance of our mechanism is also quite minimal.

We use the following lemmas in the analysis.

**Lemma 4.** *For any $j,k$, $\mathbb{P}\left(a'_{j,h_j(k)} = 1\right) = \frac{a_{j,k}+\sigma_j}{\omega_j}$.*

*Proof.* We have

$$\Pr\left[a'_{j,h_j(k)} = 1 \mid a_{j,k} = 0\right] = p'_{\mathrm{a},j} = \frac{\sigma_j}{\omega_j}.$$

On the other hand,

$$\Pr\left[a'_{j,h_j(k)} = 1 \mid a_{j,k} = 1\right] = p'_{\mathrm{p},j} = \frac{1+\sigma_j}{\omega_j}.$$

$\square$

In Lemma 2 and later, we work under the assumption that $d_i < m \approx n/s$. This assumption is reasonable for a large class of graphs as the maximum degree can often be inherently limited by physical or practical constraints. For instance, if we consider a graph representing physical interactions over the past two weeks–relevant in contexts such as epidemic modeling–the number of interactions any individual can have is bounded and cannot approach the size of the global population. Another example is the case of Facebook that imposes a hard cap of 5,000 friends per user Chandler (2012). Additionally, given that the objective of this article is to mitigate the increase in communication overhead caused by privacy measures, this assumption aligns well with our intended use-case. Notably, should $d_i$ exceed $m$, the communication cost associated with our method actually becomes lower than that of the corresponding non-private algorithm. To better under the consequence on the accuracy when that assumption is violated, we conducted experiments on power-law graphs in Section 7.1.5.

**Lemma 5.** *For all $i$ such that $d_i < m$, $\omega_i = O\left(\frac{s}{e^{\varepsilon_1}-1}\right)$ and $\sigma_i = O\left(\frac{1}{e^{\varepsilon_1}-1}\right)$.*

*Proof.* From the relation $\varepsilon' = \ln(1 + s(e^{\varepsilon_1} - 1))$, it follows that

$$e^{\varepsilon'} - 1 = s(e^{\varepsilon_1} - 1).$$

Therefore, we obtain

$$\omega_i = \frac{2 + s(e^{\varepsilon_1} - 1)}{s(e^{\varepsilon_1} - 1)} \cdot \frac{ms - 1}{m - 1} = O\left(\frac{s}{e^{\varepsilon_1} - 1}\right).$$

Moreover, under the assumption that $d_i < m$, we have

$$\sigma_i \le \frac{1}{e^{\varepsilon'} - 1} \cdot \frac{ms - 1}{m - 1} + 1 = \frac{1}{s(e^{\varepsilon_1} - 1)} \cdot \frac{ms - 1}{m - 1} + 1 \le 1 + \frac{m}{m - 1} \cdot \frac{1}{e^{\varepsilon_1} - 1} = O\left(\frac{1}{e^{\varepsilon_1} - 1}\right).$$

$\square$

We consider the variance of the variable $\tilde{a}_{j,h_j(k)}$ in the next theorem.

**Theorem 4.** *For all $j$ such that $d_j < m$, the variance of $\tilde{a}_{j,h_j(k)}$ is $O\left(\frac{s}{(e^{\varepsilon_1}-1)^2} + \frac{s^2}{n^2 \varepsilon_0^2}\right)$.*

*Proof.* We begin by analyzing the variance of the binary variable $a'_{j,h_j(k)}$. Since it takes values in $\{0, 1\}$, we apply Lemma 4 to obtain

$$\text{Var}\left(a'_{j,h_j(k)}\right) = \Pr\left[a'_{j,h_j(k)} = 1\right] \cdot \left(1 - \Pr\left[a'_{j,h_j(k)} = 1\right]\right) = \left(\frac{a_{j,k} + \sigma_j}{\omega_j}\right) \cdot \left(1 - \frac{a_{j,k} + \sigma_j}{\omega_j}\right) \le \frac{1 + \sigma_j}{\omega_j}.$$

Next, using Lemma 5, we bound the variance of the estimator $\tilde{a}_{j,h_j(k)}$ as follows:

$$\text{Var}\left(\tilde{a}_{j,h_j(k)}\right) = \text{Var}\left(\omega_j a'_{j,h_j(k)} + \tilde{\sigma}_i\right) = \omega_j^2 \text{Var}\left(a'_{j,h_j(k)}\right) + \text{Var}\left(\tilde{\sigma}_i\right) \le \omega_j(1 + \sigma_j) + \left(\frac{s-1}{ms-s}\right)^2 \cdot \frac{2}{\varepsilon_0^2}.$$

Therefore, we conclude, using the assumption that for all $j$, $d_j < m$ and Lemma 5, that

$$\text{Var}\left(\tilde{a}_{j,h_j(k)}\right) = O\left(\frac{s}{(e^{\varepsilon_1} - 1)^2} + \frac{s^2}{n^2 \varepsilon_0^2}\right).$$

$\square$

It follows from Theorem 5 and subsequent theorems that the variance component associated with $\varepsilon_1$ dominates the component associated with $\varepsilon_0$. The $\varepsilon_0$-dependent term scales as $1/n^2$ and is therefore negligible on large networks. Under a fixed privacy budget $\varepsilon = \varepsilon_0 + \varepsilon_1 + \varepsilon_2$, this suggests setting $\varepsilon_0 \ll \varepsilon_1$ to minimize variance. On the other hand, the sensitivity of the counting step—and the error introduced at Line 5 (Laplace or smooth sensitivity)—can be large for certain networks; accordingly, we recommend choosing $\varepsilon_1 \approx \varepsilon_2$.

In Section 5.5, we will elaborate that $\tilde{a}_{j,h_j(k)}$ can cut the communication cost by a factor of $s^2$ in contrast to using the direct randomized response results $a'_{j,k}$. Given that the variance of $a'_{j,k}$ is known to be $\Theta\left(\frac{1}{(e^{\varepsilon_1}-1)^2}\right)$, the previous theorem suggests that, while we manage to decrease the communication cost by a factor of $s^2$, this is at the expense of amplifying the variance by a factor of $s$. In the upcoming section, we will demonstrate that, for a set communication cost, this mechanism yields a considerably reduced variance.

### 5.4 Covariance of Variables

In situations where the goal is to sum up specific estimators, like in subgraph counting, a covariance emerges between these estimators. The size of this covariance is addressed in the subsequent theorem.

**Theorem 5.** *Assume $d_i < m$ for all $i$. The covariance of the estimators $\tilde{a}_{j,h_j(k)}$ and $\tilde{a}_{j',h_{j'}(k')}$ is in $\mathcal{O}\left(\frac{s^2}{n(e^{\varepsilon_1}-1)^2}\right)$ when $j = j'$ and $k, k' < j$. Otherwise, the covariance of the two estimators is zero.*

*Proof.* When $j \neq j'$, the two estimators are computed from disjoint sets of random variables and are therefore independent.

When $j = j'$, the estimators $\tilde{a}_{j,h_j(k)}$ and $\tilde{a}_{j,h_j(k')}$ are identical if $k$ and $k'$ fall into the same bin, i.e., $h_j(k) = h_j(k')$. We denote this event by $C_{k,k'}^{(j)}$, which occurs with probability

$$\Pr\left(C_{k,k'}^{(j)}\right) = \frac{s-1}{ms-1}.$$

As a result, for $k, k' < j$,

$$\mathrm{Cov}\left(\tilde{a}_{j,h_j(k)}, \tilde{a}_{j,h_j(k')}\right) = \omega_j^2 \cdot \mathrm{Cov}\left(a'_{j,h_j(k)}, a'_{j,h_j(k')}\right) = \omega_j^2 \left(\mathbb{E}\left[a'_{j,h_j(k)} a'_{j,h_j(k')}\right] - \mathbb{E}\left[a'_{j,h_j(k)}\right]\mathbb{E}\left[a'_{j,h_j(k')}\right]\right).$$

We consider the conditional expectations $\mathbb{E}\left[a'_{j,h_j(k)} a'_{j,h_j(k')} \mid C_{k,k'}^{(j)}\right]$ and $\mathbb{E}\left[a'_{j,h_j(k)} a'_{j,h_j(k')} \mid \overline{C_{k,k'}^{(j)}}\right]$.

Recall from the previous section that

$$\Pr\left[a_{j,h_j(k)} = 1\right] = \frac{s-1}{s} \cdot \frac{d_j}{ms-1} + a_{j,k}\left(\frac{1}{s} - \frac{1}{ms-1}\right).$$

Multiplying both sides by $\frac{ms-1}{ms-2}$ yields

$$\frac{ms-1}{ms-2} \cdot \Pr\left[a_{j,h_j(k)} = 1\right] = \frac{s-1}{s} \cdot \frac{d_j}{ms-2} + a_{j,k}\left(\frac{ms-1}{s(ms-2)} - \frac{1}{ms-2}\right) \geq \frac{s-1}{s} \cdot \frac{d_j - a_{j,k}}{ms-2} + \frac{a_{j,k}}{s}.$$

Let $p_k = \Pr\left[a_{j,h_j(k)} = 1 \mid \overline{C_{k,k'}^{(j)}}\right]$. Then,

$$p_k = \frac{a_{j,k}}{s} + \frac{s-1}{s} \cdot \frac{d_j - a_{j,k} - a_{j,k'}}{ms-2} \leq \frac{ms-1}{ms-2} \cdot \Pr\left[a_{j,h_j(k)} = 1\right].$$

It follows that

$$\Pr\left[a'_{j,h_j(k)} = 1 \text{ and } a'_{j,h_j(k')} = 1 \mid \overline{C_{k,k'}^{(j)}}\right] = \left(p_k \cdot \frac{e^{\varepsilon'}-1}{e^{\varepsilon'}+1} + \frac{1-p_k}{e^{\varepsilon'}+1}\right) \times \left(p_{k'} \cdot \frac{e^{\varepsilon'}-1}{e^{\varepsilon'}+1} + \frac{1-p_{k'}}{e^{\varepsilon'}+1}\right)$$

$$\leq \left(\frac{ms-1}{ms-2}\right)^2 \mathbb{E}\left[a'_{j,h_j(k)}\right] \mathbb{E}\left[a'_{j,h_j(k')}\right].$$

Hence, for $s \geq 3$ (which is always satisfied in practice),

$$\Pr\left[\overline{C_{k,k'}^{(j)}}\right] \cdot \mathbb{E}\left[a'_{j,h_j(k)} a'_{j,h_j(k')} \mid \overline{C_{k,k'}^{(j)}}\right] - \mathbb{E}\left[a'_{j,h_j(k)}\right]\mathbb{E}\left[a'_{j,h_j(k')}\right] \leq 0.$$

Thus,

$$\mathrm{Cov}\left(\tilde{a}_{j,h_j(k)}, \tilde{a}_{j,h_j(k')}\right) \leq \omega_j^2 \cdot \mathbb{E}\left[a'_{j,h_j(k)} a'_{j,h_j(k')} \mid C_{k,k'}^{(j)}\right] \cdot \Pr\left(C_{k,k'}^{(j)}\right).$$

Let $p'_k = \Pr\left[a_{j,h_j(k)} = 1 \mid C^{(j)}_{k,k'}\right]$. Then,

$$p'_k = \frac{a_{j,k} + a_{j,k'}}{s} + \frac{s-2}{s} \cdot \frac{d_j - a_{j,k} - a_{j,k'}}{ms-2} \leq \Pr\left[a_{j,h_j(k)} = 1\right] + \Pr\left[a_{j,h_j(k')} = 1\right].$$

It follows that

$$\mathbb{E}\left[a'_{j,h_j(k)} a'_{j,h_j(k')} \mid C^{(j)}_{k,k'}\right] \leq \mathbb{E}\left[a'_{j,h_j(k)}\right] + \mathbb{E}\left[a'_{j,h_j(k')}\right].$$

Using Lemma 4 and $\Pr\left(C^{(j)}_{k,k'}\right) = \frac{s-1}{ms-1}$, we obtain

$$\mathrm{Cov}\left(\tilde{a}_{j,h_j(k)}, \tilde{a}_{j,h_j(k')}\right) \leq 2 \cdot \frac{s-1}{ms-1} \cdot \omega_j (1+\sigma_j) = O\left(\frac{s^2}{n(e^{\varepsilon_1}-1)^2}\right).$$

$\square$

When $s \ll n$, the covariance in the previous theorem is significantly smaller than the variance calculated in Theorem 4. While the variance of the summation includes more covariance terms than variance terms, we will demonstrate in the next section that the contribution from the covariance terms is not substantially larger than the that of the variance terms in Theorem 4.

## 5.5 Communication Cost

The final aspect we will examine regarding the mechanism's efficiency is the communication needed for graph aggregation and distribution to each user. We will particularly look at two metrics: the "download cost" (the average number of bits the server transmits to each user) and the "upload cost" (the average number of bits each user sends to the server). Our analysis will offer approximate values under the assumption that the graph is sparse, meaning the number of edges ($|E|$) is much less than the square of the number of vertices ($|V|^2$).

From our discussion in the previous section, it is evident that the majority of the upload cost arises from the randomized response step, while the download cost primarily stems from the counting step. Therefore, in this analysis, we will disregard the costs associated with the other steps.

**Theorem 6.** *The upload cost of user $v_i$ in our mechanism is $O\left(\left(\frac{d_i}{s} + \frac{n}{s^2(e^{\varepsilon_1}-1)}\right)\log\frac{n}{s}\right).$*

*Proof.* Recall that, in the randomized response step, each user $v_i$ submits a binary vector $[a'_{i,c_1}, \ldots, a'_{i,c_m}] \in \{0,1\}^m$, where $m \approx n/s$. To reduce communication cost, the user may instead transmit only the index set $\{t : a'_{i,c_t} = 1\} \subseteq \{1,\ldots,m\}$. If $\ell$ entries in the vector are equal to one, the communication cost is $\ell \cdot \log m \approx \ell \cdot \log(n/s)$.

We now analyze the expected value of $\ell$. Since each $a_{i,j}$ is sampled into the vector with probability $1/s$, and noting that $e^{\varepsilon'} - 1 = s(e^{\varepsilon_1} - 1)$, we obtain:

$$\mathbb{E}[\ell] = \left(\frac{e^{\varepsilon'}}{1+e^{\varepsilon'}} \cdot d_i + \frac{1}{1+e^{\varepsilon'}} \cdot (n-d_i)\right) \cdot \frac{1}{s} \leq \frac{d_i}{s} + \frac{1}{s^2(e^{\varepsilon_1}-1)} \cdot (n-d_i) = O\left(\frac{d_i}{s} + \frac{n}{s^2(e^{\varepsilon_1}-1)}\right).$$

$\square$

Next, we consider the download cost for our mechanism.

**Corollary 1.** *The download cost for all users is $O\left(\log\frac{n}{s}\left(\frac{|E|}{s} + \frac{n^2}{s^2(e^{\varepsilon_1}-1)}\right)\right).$*

*Proof.* Since each user must download information from all other users, the total download cost is

$$\sum_i O\left(\log\left(\frac{n}{s}\right)\left(\frac{d_i}{s} + \frac{n}{s^2(e^{\varepsilon_1}-1)}\right)\right) = O\left(\log\left(\frac{n}{s}\right)\left(\frac{|E|}{s} + \frac{n^2}{s^2(e^{\varepsilon_1}-1)}\right)\right),$$

where $|E| = \sum_i d_i$ is the total number of edges. $\square$

When $|E| = O(n)$, the upload cost is $O\left(\frac{n}{s^2} \log n\right)$ and the download cost is $O\left(\frac{n^2}{s^2} \log n\right)$. Recall that the communication cost in the two-step mechanism is $\Theta(n^2)$. Therefore, our technique reduces the communication cost by a factor of $s^2$.

## 6 Use Case: Triangle Counting

While our GroupRR mechanism is applicable to the counting of various subgraphs, the majority of prior work—such as Imola et al. (2021; 2022a); Eden et al. (2023)—has focused specifically on triangle counting under local differential privacy. To facilitate direct comparison with these existing approaches, we also focus on triangle counting in this section.

In Subsection 6.1, we analyze the estimation loss introduced by our mechanism, excluding the effect of Line 5 in the counting step of Algorithm 3, where noise is added via either the smooth sensitivity or local Laplace mechanism. In Subsection 6.2, we propose an algorithmic refinement aimed at further reducing this estimation loss for triangle counting. Finally, Subsections 6.3 and 6.4 are devoted to analyzing the noise contribution introduced in the final step of the algorithm.

### 6.1 Sensitivity and Loss of Our Mechanism

In the counting step of our mechanism, each user $v_i$ estimates $|\Delta_i|$ when $\Delta_i = \{(v_i, v_j, v_k) : k < j < i, \{v_i, v_j\}, \{v_i, v_k\}, \{v_j, v_k\} \in E\}$. The number of triangles in the graph $G$, denoted by $|\Delta_G|$ is then the sum of all $|\Delta_i|$. The estimation of $|\Delta_i|$ in our mechanism is $|\tilde{\Delta}_i| := \sum\limits_{j,k:\{i,j\},\{i,k\}\in E} \tilde{a}_{j,h_j(k)}$ added with the Laplacian noise when the variable $\tilde{a}_{j,h_j(k)}$ is defined in the previous section. The variance of our mechanism without the noise in the final step of our mechanism is shown in the subsequent theorem.

**Theorem 7.** *Let $S_2$, $C_4$, and $W_4$ denote the number of two-stars, four-cycles, and walks of length four in the graph $G$, respectively. Then, the variance of the estimator $\sum_i \sum\limits_{j,k:\{i,j\},\{i,k\}\in E} \tilde{a}_{j,h_j(k)}$ is bounded by*

$$O\left(\frac{s}{(e^{\varepsilon_1} - 1)^2} \cdot \left(S_2 + C_4 + \frac{s}{n} \cdot W_4\right)\right).$$

*Proof.* The variance of the estimator $\sum_i \sum\limits_{j,k:\{i,j\},\{i,k\}\in E} \tilde{a}_{j,h_j(k)}$ can be decomposed into the following three components:

1. **Variance of individual terms:** From Theorem 4, we have $\text{Var}\left(\tilde{a}_{j,h_j(k)}\right) = O\left(\frac{s}{(e^{\varepsilon_1}-1)^2}\right)$. As there are $S_2$ such terms (one for each two-star), the total contribution from the variances is

$$O\left(\frac{s \cdot S_2}{(e^{\varepsilon_1} - 1)^2}\right).$$

2. **Covariance due to repeated terms:** A term $\tilde{a}_{j,h_j(k)}$ may appear multiple times in the summation if there exist $i, i' > j > k$ such that $\{i,j\}, \{i,k\}, \{i',j\}, \{i',k\} \in E$. Such configurations correspond to four-cycles, of which there are at most $C_4$. From Theorem 4, each such repeated term contributes $O\left(\frac{s}{(e^{\varepsilon_1}-1)^2}\right)$ to the total covariance, yielding a contribution bounded by

$$O\left(\frac{s \cdot C_4}{(e^{\varepsilon_1} - 1)^2}\right).$$

3. **Covariance between distinct terms:** Dependencies may also arise between distinct terms $\tilde{a}_{j,h_j(k)}$ and $\tilde{a}_{j,h_j(k')}$ if there exist $i > j, k$ and $i' > j, k'$ such that the edge sets $\{i,j\}, \{i,k\}, \{i',j\}, \{i',k'\} \in$

$E$. These configurations correspond to walks of length four involving shared center nodes. There are at most $W_4$ such configurations, and each contributes $O\left(\frac{s^2}{n(e^{\varepsilon_1}-1)^2}\right)$ to the total covariance, resulting in an overall contribution of

$$O\left(\frac{s^2 \cdot W_4}{n(e^{\varepsilon_1}-1)^2}\right).$$

Combining all three components, we conclude that the total variance is

$$O\left(\frac{s}{(e^{\varepsilon_1}-1)^2} \cdot \left(S_2 + C_4 + \frac{s}{n} \cdot W_4\right)\right).$$

$\square$

## 6.2 Further Optimization for Triangle Counting: Central Server Sampling

We believe that the variance detailed in Theorem 7 is comparatively minimal. Nevertheless, specifically for triangle counting, this variance can be further reduced through a method we have termed "central server sampling." The specifics of this technique will be outlined in this subsection.

### 6.2.1 Modification

Remember that through the GroupRR mechanism, the server acquires the set $\bar{\xi}_i = \{t \mid a'_{i,c_t} = 1\}$ from user $v_i$ following the group randomized response step. This set is then shared with all users at the start of the counting step. As identified in the prior subsection, this approach not only leads to significant communication costs but also contributes to a substantial covariance in triangle counting. To address these issues, we suggest generating subsets $\bar{\xi}_i^{(1)}, \ldots, \bar{\xi}_i^{(n)}$ from $\bar{\xi}_i$ independently, and then distributing the subset $\bar{\xi}_i^{(j)}$ to user $v_j$. An element $t$ in $\bar{\xi}_i$ is independently selected for inclusion in $\bar{\xi}_i^{(j)}$ with a constant probability, which we denote as $\mu_c$. The central server sampling is described in Algorithm 4.

---

**1** | **Function** `CentralSampling`
| **Input:** For all user $j$, the grouped and obfuscated adjacency list $(a'_{j,c_1}, \ldots, a'_{j,c_m})$ obtained from the randomized response step, central server sampling probability $\mu_C$
| **Output:** For all user $i, j$, the grouped and obfuscated adjacency list for user $j$, which user $i$ will use for counting the number of triangles in the counting step, $(a^{(i)}_{j,c_1} \ldots, a^{(i)}_{j,c_m})$.
**2** | [**Server**] For each $i, j, t$, $a^{(i)}_{j,c_t} = 0$ if $a'_{j,c_t} = 0$. If $a'_{j,c_t} = 1$, $a^{(i)}_{j,c_t} = 1$ with probability $\mu_c$, and $a^{(i)}_{j,c_t} = 0$ with probability $1 - \mu_c$.;
**3** | [**Server**] Send $(a^{(i)}_{j,c_1} \ldots, a^{(i)}_{j,c_m})$ to user $i$.

**Algorithm 4:** Central Server Sampling: This process is executed between Step 3 and 4 of the GroupRR mechanism

---

Let $a^{(i)}_{j,h_j(k)} \in \{0,1\}$ is a random variable indicating if $h_j(k) \in \bar{\xi}_j^{(i)}$. The user $v_i$ believe that there is an edge $\{v_j, v_k\}$ in the graph $G$ if $a^{(i)}_{j,h_j(k)} = 1$. We have that $\mathbb{E}\left[a^{(i)}_{j,h_j(k)}\right] = \mu_c \cdot \mathbb{E}\left[a'_{j,h_j(k)}\right]$ for all $i, j, k$.

Recall the variables $\omega_i$ and $\tilde{\sigma}_i$ defined in the previous section. We define $\tilde{a}^{(i)}_{j,h_j(k)} = \frac{\omega_i}{\mu_c} a^{(i)}_{j,h_j(k)} - \tilde{\sigma}_i$.

**Theorem 8.** *The variable $\tilde{a}^{(i)}_{j,h_j(k)}$ is an unbiased estimation of $a_{j,k}$.*

*Proof.* By the definitions and Theorem 3, we have

$$\mathbb{E}[\tilde{a}^{(i)}_{j,h_j(k)}] = \frac{\omega_i}{\mu_c} \cdot \mathbb{E}[a^{(i)}_{j,h_j(k)}] - \mathbb{E}[\tilde{\sigma}_i] = \omega_i \cdot \mathbb{E}[a'_{j,h_j(k)}] - \mathbb{E}[\tilde{\sigma}_i] = a_{j,k}.$$

$\square$

From the previous theorem, our unbiased estimation of the number of triangles under this modification is then $|\tilde{\Delta}_i| = \sum\limits_{j,k:\{v_i,v_j\},\{v_i,v_k\}\in E} \tilde{a}^{(i)}_{j,h_j(k)}$ added by the Laplacian noise or the noise from the smooth sensitivity mechanism.

### 6.2.2 Variance of the Modified Mechanism

We first consider the variance of $\tilde{a}^{(i)}_{j,h_j(k)}$ in the subsequent theorem.

**Theorem 9.** $Var\left(\tilde{a}^{(i)}_{j,h_j(k)}\right) = O\left(\frac{s}{\mu_c(e^{\varepsilon_1}-1)^2} + \frac{s^2}{n^2\varepsilon_0^2}\right).$

*Proof.* Since $a^{(i)}_{j,h_j(k)}$ is a binomial random variable, Lemma 4 implies

$$
\begin{aligned}
\mathrm{Var}\left(a^{(i)}_{j,h_j(k)}\right) &= \Pr\left[a^{(i)}_{j,h_j(k)}=1\right]\cdot\left(1-\Pr\left[a^{(i)}_{j,h_j(k)}=1\right]\right)\\
&= \mu_c\cdot\Pr\left[a'_{j,h_j(k)}=1\right]\cdot\left(1-\mu_c\cdot\Pr\left[a'_{j,h_j(k)}=1\right]\right)\\
&\leq \mu_c\cdot\Pr\left[a'_{j,h_j(k)}=1\right]\leq \mu_c\cdot\frac{1+\sigma_j}{\omega_j}.
\end{aligned}
$$

Then, by Lemma 5, we have

$$
\mathrm{Var}\left(\tilde{a}^{(i)}_{j,h_j(k)}\right) = \frac{\omega_j^2}{\mu_c^2}\cdot\mathrm{Var}\left(a^{(i)}_{j,h_j(k)}\right) + \mathrm{Var}\left(\tilde{\sigma}_i\right) \leq \frac{1}{\mu_c}\cdot\frac{1+\sigma_j}{\omega_j} + \frac{s^2}{n^2\varepsilon_0^2}.
$$

Therefore,

$$
\mathrm{Var}\left(\tilde{a}^{(i)}_{j,h_j(k)}\right) = O\left(\frac{s}{\mu_c(e^{\varepsilon_1}-1)^2} + \frac{s^2}{n^2\varepsilon_0^2}\right).
$$

$\square$

At first glance, when contrasting this outcome with the findings in Theorem 4, it might appear that employing central server sampling amplifies the variance of the estimations, potentially diminishing their utility. However, it is important to recognize that for a communication cost reduction by $1/\mu_c$ times. Additionally, in the same way as the four-cycle trick in Section 3, the central server sampling can reduce the covariance stated in Theorem 7. In particular, it significantly reduces the covariance for the graph with a lot of walks with length four (or graph with large number of $W_4$).

**Theorem 10.** *The covariance of $\tilde{a}^{(i)}_{j,h_j(k)}$ and $\tilde{a}^{(i')}_{j',h'_j(k')}$ is $O\left(s/(e^{\varepsilon_1}-1)^2\right)$ when $j=j'$, $i\neq i'$ and $k=k'$, $O\left(s^2/(n(e^{\varepsilon_1}-1)^2)\right)$ when $j=j'$, $i\neq i'$ and $k,k'<j$, and $O\left(s^2/(\mu_c n(e^{\varepsilon_1}-1)^2)\right)$ when $j=j'$, $i=i'$ and $k,k'<j$. It is equal to zero in other cases.*

*Proof.* When $j\neq j'$, the two estimators are based on disjoint sets of variables, and the server sampling step is conducted independently for each group. Therefore, the estimators are independent.

When $j=j'$, $i\neq i'$, and $k,k'<j$, the sampling steps for $\tilde{a}^{(i)}_{j,h_j(k)}$ and $\tilde{a}^{(i')}_{j,h_j(k')}$ are independent. Hence, by Theorem 5,

$$
\mathrm{Cov}\left(\tilde{a}^{(i)}_{j,h_j(k)},\tilde{a}^{(i')}_{j,h_j(k')}\right) \leq \mathrm{Cov}\left(\tilde{a}_{j,h_j(k)},\tilde{a}_{j,h_j(k')}\right) = O\left(\frac{s^2}{n(e^{\varepsilon_1}-1)^2}\right).
$$

Similarly, when $j=j'$, $i\neq i'$, and $k=k'$, Theorem 4 gives

$$
\mathrm{Cov}\left(\tilde{a}^{(i)}_{j,h_j(k)},\tilde{a}^{(i')}_{j,h_j(k)}\right) \leq \mathrm{Var}\left(\tilde{a}_{j,h_j(k)}\right) = O\left(\frac{s}{(e^{\varepsilon_1}-1)^2}\right).
$$

Next, consider the case when $j = j'$, $i = i'$, and $k, k' < j$. We distinguish two subcases depending on whether $h_j(k) = h_j(k')$. Let $C_{k,k'}^{(j)}$ denote the event that $h_j(k) = h_j(k')$. By Theorem 9,

$$\mathbb{E}\left[\tilde{a}_{j,h_j(k)}^{(i)}\tilde{a}_{j,h_j(k')}^{(i)} \mid C_{k,k'}^{(j)}\right] = \mathrm{Var}\left(\tilde{a}_{j,h_j(k)}^{(i)}\right) = O\left(\frac{s}{\mu_c(e^{\varepsilon_1} - 1)^2}\right).$$

In the case $h_j(k) \neq h_j(k')$, we obtain from the proof of Theorem 5 that

$$\mathbb{P}\left(\overline{C_{k,k'}^{(j)}}\right) \cdot \mathbb{E}\left[a_{j,h_j(k)}^{(i)}a_{j,h_j(k')}^{(i)} \mid \overline{C_{k,k'}^{(j)}}\right] - \mathbb{E}\left[a_{j,h_j(k)}^{(i)}\right]\mathbb{E}\left[a_{j,h_j(k')}^{(i)}\right] \leq 0.$$

Therefore,

$$\mathrm{Cov}\left(\tilde{a}_{j,h_j(k)}^{(i)}, \tilde{a}_{j,h_j(k')}^{(i)}\right) \leq \mathbb{P}\left(C_{k,k'}^{(j)}\right) \cdot \mathbb{E}\left[\tilde{a}_{j,h_j(k)}^{(i)}\tilde{a}_{j,h_j(k')}^{(i)} \mid C_{k,k'}^{(j)}\right] = O\left(\frac{s^2}{\mu_c n(e^{\varepsilon_1} - 1)^2}\right).$$

$\square$

Using the previous two theorems, we will quantify the variance of the triangle counting algorithm performed using GroupRR and central server sampling. We describe in the next theorem this variance prior to the addition of the Laplacian noise in the final step of the mechanism.

**Theorem 11.** *Let $S_2$, $S_3$, $C_4$, and $W_4$ be the number of two-star, three-star, four-cycle, and the walk with length four in $G$. The variance of our estimator for triangle counting using the GroupRR and central server sampling technique, denoted by $\sum_i \left(\sum_{j,k:\{v_i,v_j\},\{v_i,v_k\}\in E} \tilde{a}_{j,h_j(k)}^{(i)}\right)$, is*
$\mathcal{O}\left(\frac{s}{\mu_c(e^{\varepsilon_1}-1)^2}\left(S_2 + \mu_c C_4 + \frac{\mu_c s}{n}W_4 + \frac{s}{n}S_3\right)\right).$

*Proof.* The variance of

$$\sum_i \left(\sum_{\substack{j,k: \\ \{v_i,v_j\},\{v_i,v_k\}\in E}} \tilde{a}_{j,h_j(k)}^{(i)}\right)$$

can be decomposed into four components:

1. **Variance of individual terms:**

$$\sum_i \sum_{\substack{j,k: \\ \{v_i,v_j\},\{v_i,v_k\}\in E}} \mathrm{Var}\left(\tilde{a}_{j,h_j(k)}^{(i)}\right)$$

   By Theorem 9, we have

$$\mathrm{Var}\left(\tilde{a}_{j,h_j(k)}^{(i)}\right) = O\left(\frac{s}{\mu_c(e^{\varepsilon_1} - 1)^2}\right).$$

   Since there are at most $S_2$ such terms in total, the overall contribution is $O\left(\frac{s \cdot S_2}{\mu_c(e^{\varepsilon_1}-1)^2}\right)$.

2. **Covariance from repeated index pairs $\{j,k\}$:** This arises from the covariance between $\tilde{a}_{j,h_j(k)}^{(i)}$ and $\tilde{a}_{j,h_j(k)}^{(i')}$ for $i, i' > j > k$ such that $\{v_i,v_j\}$, $\{v_i,v_k\}$, $\{v_{i'},v_j\}$, and $\{v_{i'},v_k\}$ are all in $E$. By Theorem 10, the covariance is bounded by $O\left(\frac{s}{(e^{\varepsilon_1}-1)^2}\right)$, and the number of such co-occurrences is at most $C_4$. Hence, this part contributes $O\left(\frac{s \cdot C_4}{(e^{\varepsilon_1}-1)^2}\right)$.

3. **Covariance across different** $i \neq i'$**:** This refers to terms $\tilde{a}^{(i)}_{j,h_j(k)}$ and $\tilde{a}^{(i')}_{j,h_j(k')}$ where $i \neq i'$, and there exist $i > j, k$ and $i' > j, k'$ such that $\{v_i, v_j\}, \{v_i, v_k\}, \{v_{i'}, v_j\}, \{v_{i'}, v_{k'}\} \in E$. Each such occurrence contributes at most $O\left(\frac{s^2}{n(e^{\varepsilon 1}-1)^2}\right)$, and the total number of such cases is bounded by $W_4$, leading to a total contribution of $O\left(\frac{s^2 \cdot W_4}{n(e^{\varepsilon 1}-1)^2}\right)$.

4. **Covariance within the same** $i$**:** This arises from pairs $\tilde{a}^{(i)}_{j,h_j(k)}$ and $\tilde{a}^{(i)}_{j,h_j(k')}$ for fixed $i$ and $j$, where $i > j$ and the edge set contains $\{v_i, v_j\}, \{v_i, v_k\}, \{v_i, v_{k'}\} \in E$. Each such pair contributes at most $O\left(\frac{s^2}{\mu_c n(e^{\varepsilon 1}-1)^2}\right)$, and there are at most $S_3$ such cases, giving a total contribution of $O\left(\frac{s^2 \cdot S_3}{\mu_c n(e^{\varepsilon 1}-1)^2}\right)$.

$\square$

Let us compare the variance derived from the GroupRR as outlined in Theorem 7 with that obtained from GroupRR combined with central server sampling, as detailed in Theorem 11, particularly for graphs with numerous walks of length four. In such scenarios, the term $W_4$ becomes the dominant factor in the variances of both methods. We find that the constant associated with this dominant term is identical in both variances. This observation suggests that while achieving a similar level of variance, we can also reduce the download cost by a factor of $1/\mu_c$.

## 6.3 Smooth Sensitivity for Triangle Counting

We calculate the sensitivity at distance $k$ (defined in Definition 6) for the triangle counting. First, let us consider the local sensitivity (defined in Definition 5) for the adjacency vector $a' \in \{0, 1\}^n$ (denoted by $LS_f(a')$). As previously discussed, the maximum change in the count $\sum_{j,k:a'_j=a'_k=1} \tilde{a}^{(i)}_{j,h_j(k)}$ when one bit of $a'$ is updated is no more than

$$UB_{\text{LS}}(a') = \max_j \left( \sum_{k:a'_k=1} \tilde{a}^{(i)}_{j,h_j(k)} + \sum_{k:a'_k=1} \tilde{a}^{(i)}_{k,h_k(j)} \right).$$

When $a_i$ is the adjacent vector of user $i$ and $d_i$ is the number of ones in the vector, the number of ones in the vector $a$ with $|a - a_i| \leq k$ is not more than $UB_{\text{LS}}(a_i) + k \cdot \omega_i$. Hence, the sensitivity at $k$ of the vector $a_i$, denoted by $A^{(k)}(a_i)$ is no more than $O\left(UB_{\text{LS}}(a_i) + \frac{k \cdot s}{e^{\varepsilon 1}+1}\right)$. Combining with Theorem 11, we obtain that:

**Theorem 12.** *When using the smooth sensitivity mechanism, the variance of our estimator is* $O(\frac{1}{\varepsilon_2^2} \sum_{i=1}^n \left(UB_{LS}(a_i) + \frac{s}{\varepsilon_2(e^{\varepsilon 1}+1)}\right)^2 + \frac{s}{\mu_c(e^{\varepsilon 1}-1)^2}\left(S_2 + \mu_c C_4 + \frac{\mu_c s}{n}W_4 + \frac{s}{n}S_3\right)).$

## 6.4 Clipping for Triangle Counting

Although the smooth sensitivity mechanism discussed in the previous subsection provides accurate results, calculating the sensitivity $UB_{\text{LS}}(a_i)$ can be time-consuming in large networks. In such cases, we opt for the local Laplacian mechanism (Definition 3).

Let $f : \{0, 1\}^n \to \mathbb{R}$ be defined as $f(a') = \sum_{j,k:a'_j=a'_k=1} \tilde{a}^{(i)}_{j,h_j(k)}$, representing the value we intend to publish for user $i$. Recall that the magnitude of the Laplacian noise added by the mechanism is determined by the global sensitivity $\Delta_f = \max_{a',a'':|a'-a''|=1} (f(a') - f(a''))$. Given that $\tilde{a}^{(i)}_{j,h_j(k)} \leq \omega_i$, we find that $\Delta_f \leq n \cdot \omega_i$.

We can use $n \cdot \omega_i$ as the magnitude of the Laplacian noise. However, $n \cdot \omega_i$ is to large that the noise can dominate the information we intend to publish. To have a better bound for the sensitivity, we employ the ideas of double clipping in Imola et al. (2022a). Although we use some ideas from the paper, as our mechanism is different from theirs, we attain a better sensitivity by a different mathematical analysis. Our double counting algorithm is as follows:

1. **Degree Clipping:** Recall that $\tilde{d}_i$ is the estimation for the degree of $v_i$ published in the first step of our mechanism. Let $\hat{d}_i = \tilde{d}_i + \varepsilon_0 \ln \frac{2}{\beta}$[1].

2. **Noisy Triangle Clipping:** Let Var represent the maximum variance of a single edge estimator, Cov the maximum covariance between two edge estimators, and define $b_i = \hat{d}_i + \sqrt{\frac{2}{\beta}(\hat{d}_i \cdot \mathrm{Var} + \hat{d}_i^2 \cdot \mathrm{Cov})}$.
   Let $|\tilde{\Delta}_i^{(j')}| = \sum_{(j,k)\in W_i : j' \in \{j,k\}} \tilde{a}_{j,h_j(k)}^{(i)}$, where $|\tilde{\Delta}_i^{(j')}|$ denotes the contribution of node $j'$ to $|\tilde{\Delta}_i|$. During the calculation of $|\tilde{\Delta}_i|$ in Line 4 of Algorithm 4, if there exists a $j'$ such that $|\tilde{\Delta}_i^{(j')}| > b_i$, we select a pair $(j,k) \in W_i$ such that $\tilde{a}_{j,h_j(k)}^{(i)} = \omega_i - \tilde{\sigma}_i$ and update the value to 0. This process is repeated until $|\tilde{\Delta}_i^{(j')}| \le b_i$.

The clipping process described above ensures that the contribution of any edge $(i, j')$ to the publication of $|\tilde{\Delta}_i|$ does not exceed $b_i \ll n \cdot \omega_i$. As a result, the sensitivity of the publication of $|\tilde{\Delta}_i|$ is $b_i$, allowing us to set the Laplacian noise to this value.

While the clipping process allows us to have a smaller noise in the final step of our mechanism, the process can incur bias to our publication. We give a theoretical result for this clipping technique in the following theorem.

**Theorem 13.** *The clipping process incur no bias with probability at least $1 - \beta$. The variance of our estimator by the clipping process is $O(\frac{\sum_i b_i^2}{\varepsilon_2^2} + \frac{s}{\mu_c(e^{\varepsilon_1}-1)^2}\left(S_2 + \mu_c C_4 + \frac{\mu_c s}{n}W_4 + \frac{s}{n}S_3\right))$*

We use the following 2 lemmas in our analysis.

**Lemma 6.** *The probability that we remove some edges in the degree clipping is smaller than $\beta/2$.*

*Proof.* We remove edges when $d_i > \hat{d}_i = \tilde{d}_i + \varepsilon_0 \ln \frac{2}{\beta}$. This indicates that $\tilde{d}_i < d_i - \varepsilon_0 \ln \frac{2}{\beta}$, meaning the magnitude of the Laplacian noise added during the degree-sharing step is greater than $\varepsilon_0 \ln \frac{2}{\beta}$. This occurs with a probability smaller than $\beta/2$. $\qquad\square$

While we consider the degree clipping in the previous lemma, we consider the noisy triangle clipping in the subsequent lemma.

**Lemma 7.** *The probability that we update some $\tilde{a}_{j,h_j(k)}^{(i)}$ in the noisy triangle clipping is not greater than $\beta/2$.*

*Proof.* By the degree clipping and the definition of $W_i$, the number of terms in the summation

$$\sum_{(j,k)\in W_i : j' \in \{j,k\}} \tilde{a}_{j,h_j(k)}^{(i)}$$

is at most $\hat{d}_i$. From Theorem 8, we have that

$$\mathbb{E}[\tilde{a}_{j,h_j(k)}^{(i)}] = a_{j,k} \in \{0, 1\}.$$

Therefore,

$$\mathbb{E}\left[\sum_{(j,k)\in W_i : j' \in \{j,k\}} \tilde{a}_{j,h_j(k)}^{(i)}\right] \le \hat{d}_i.$$

Since the summation contains at most $\hat{d}_i$ terms, its variance can be bounded by

$$\mathrm{Var}\left(\sum_{(j,k)\in W_i : j' \in \{j,k\}} \tilde{a}_{j,h_j(k)}^{(i)}\right) \le \hat{d}_i \cdot \mathrm{Var} + \hat{d}_i^2 \cdot \mathrm{Cov}.$$

---

[1]We notice that, in Imola et al. (2022a), edges are removed to match with $\hat{d}_i$ if $d_i > \hat{d}_i$. However, that process can be skipped, because we clip the sensitivity in the noisy triangle process anyway.

Using the bounds on the expectation and variance, we conclude that

$$\Pr\left[|\tilde{\Delta}_i^{(j')}| > b_i\right] \leq \frac{\beta}{2}.$$

$\square$

We are now ready to prove Theorem 13.

*Proof of Theorem 13.* From Lemmas 6, 7, and union bound, we obtain that the clipping has no effect and incur no bias with probability at most $\beta/2$. The variance of the Laplacian noise added by the node $v_i$ is $b_i^2/\varepsilon_2^2$. Therefore, the final step of our mechanism increases the variance by $\sum_i b_i^2/\varepsilon_2^2$. $\square$

In most practical cases, we have that $b_i = O\left(\frac{d_i \cdot s}{\sqrt{\beta}(e^{\varepsilon_1}-1)}\right)$. Based on that result, in all our experiments, the failure probability $\beta$ was set to $10^{-3}$.

One might wonder why we do not set the value of $b_i$ to $\max_{j'}|\tilde{\Delta}_i^{(j')}|$, as it is smaller than the value used in the current clipping process. However, this is not feasible because $\max_{j'}|\tilde{\Delta}_i^{(j')}|$ contains sensitive information, and we cannot determine the noise parameter of our mechanism based on such private data.

In contrast, the variance and covariance required for the computation of the clipping parameter $b_i$ can be computed using $n, s, \varepsilon$ and $d_i$. With the exception of $d_i$, all those are public information. Concerning $d_i$, it can be bounded with high probability by 0 and $\hat{d}_i$ to obtain a bound on $b_i$. Note that in the rare case that $d_i > \hat{d}_i$, the clipping mechanism still guaranties the privacy protection.

## 7 Experiments

In this section, we assess our method, GroupRR, in comparison to the leading communication-constrained graph publishing technique, ARR, as detailed in Imola et al. (2022a). As Imola et al. (2022a) has proven itself to be a more accurate algorithm than the rest of the state of the art, we restrict ourself to the comparison with this algorithm. We focus on the task of counting triangles in Subsection 7.1 and the task of counting 4-cycles in Subsection 7.2. All the code used for these experiments can be accessed at the following address `https://github.com/Gericko/GroupRandomizedResponse`.

### 7.1 Triangle Counting

In this subsection, we compare the triangle counting algorithm described in earlier sections with the leading-edge algorithm named ARROne from Imola et al. (2022a)[2]. Among the various algorithms discussed in the paper, our analysis focuses on the one that utilizes the 4-cycle trick, which has been shown to yield the highest performance.

Our experiments primarily utilized the Wikipedia Article Networks dataset (Leskovec et al., 2010b;a), which contains 7,115 nodes and 100,762 edges, and the Facebook dataset (Leskovec & Mcauley, 2012), which contains 4,039 nodes and 88,234 edges. We chose these two graphs to illustrate two different types of social networks. The Facebook graph represents a network with many clusters, while the Wikipedia graph exemplifies networks centered around a few key nodes. Additionally, to prove that our methods scales to larger graph, we also used the Google+ dataset (Leskovec & Mcauley, 2012) that is constructed from Google circles. It contains 107,614 nodes and 13,673,453 edges.

---

[2]It is important to note, as mentioned in Appendix I of Imola et al. (2022a), that the original version of the algorithm underestimates the sensitivity. Consequently, we have adjusted the sensitivity calculations used in our evaluations. Specifically, in our modified version, the contribution of an edge $(v_i, v_j)$ to the count by user $v_i$ is defined as $|\{k : a_{i,k} = 1, \{v_j, v_k\} \in M_i, \text{ and } k < i\}|$ instead of $|\{k : a_{i,k} = 1, \{v_j, v_k\} \in M_i, \text{ and } j < k < i\}|$.

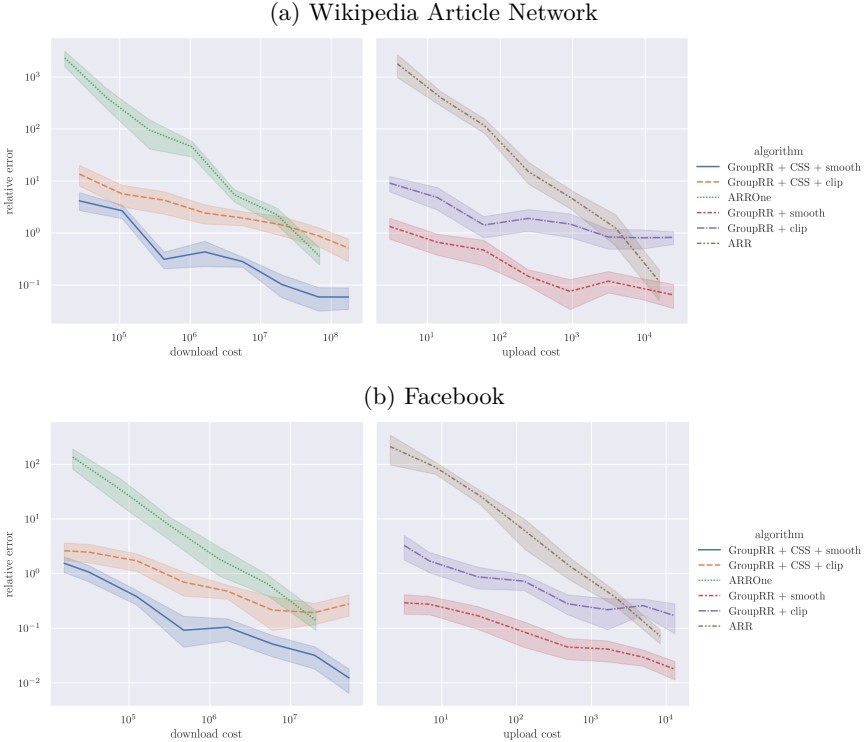

Figure 3: Comparative analysis of relative error between our algorithm and ARR across different download and upload costs applied to the Wikipedia Article Network and the Facebook graphs, assuming a privacy budget of $\varepsilon = 1$. The values of the parameter $s$ used for these experiments vary from 1 to 26 for the Wikipedia Article Network, and from 1 to 21 for Facebook, resulting in values of $\mu_c$ varying from 1 to 16,384 and from 1 to 8,192 respectively. The term 'CSS' in the legend stands for our central server sampling method

To conduct experiments on graphs of different sizes from the original, we generated extracted graphs by randomly selecting nodes to match the desired size and then examining the subgraph induced by these nodes. We have verified that selecting random subgraphs does not alter the graph topology of either graphs with respect to the subgraph counting problem.

The focus of our experiments is on assessing accuracy, measured by relative error, as we adjust various parameters. These parameters include download cost, upload cost, graph size, and privacy budget. The results presented are averages calculated from 10 separate simulation runs for each set of parameters. All experiments are comparisons between the following three methods: (1) The leading-edge method detailed in Imola et al. (2022a) (2) Our GroupRR mechanism enhanced by the central server sampling and the clipping method (Subsection 6.4) (3) Our GroupRR mechanism enhanced by the central server sampling and the smooth sensitivity (Subsection 6.3)

### 7.1.1 Error Analysis for Various Download Costs

Figure 3 illustrates that our algorithm outperforms the leading-edge algorithm across all download costs. When the download cost is minimal, the enhancement in relative error can be as substantial as a 1000-fold reduction. As the cost of communication goes down, the difference in performance between our method and the one described in Imola et al. (2022a) becomes more noticeable.

**Parameters selection for GroupRR** Recall that the download cost of the randomized response algorithm is approximately $\mathsf{c}n^2$, for some constant $\mathsf{c}$ depending on the privacy budget $\varepsilon$. Suppose our budget for the download cost is $M$; then we aim to reduce the cost by a factor of $\mu^* = \mathsf{c}n^2/M$. As discussed in the

previous section, for GROUPRR with central server sampling, this reduction factor is given by $\mu^* = \mu_c/s^2$, where $\mu_c$ and $s$ are tunable parameters in GROUPRR. To achieve the desired reduction, we set $s = (1/\mu^*)^{1/3}$ and $\mu_c = (\mu^*)^{1/3}$. Under this setting, the $S_2$ term in the error bound (as described in Theorem 11) scales as $(\mu^*)^{-2/3}$, while the $C_4$ term scales as $(\mu^*)^{-1/3}$.

**Parameters selection for ARROne**  On the other hand, in the context of ARROne, where $\mu$ represents the sampling rate of the mechanism, the reduction in download costs amounts to $\mu^2$. We hence set $\mu$ to $\sqrt{\mu^*}$. By that, the $S_2$ factor in the error term scales with $1/\mu^*$, and the $C_4$ factor scales with $1/\sqrt{\mu^*}$. This scaling behavior highlights that our algorithm proves more effective at lower communication costs compared to the method proposed in Imola et al. (2022a).

When the download cost reaches its maximum — a scenario that occurs when no sampling is implemented — our mechanism and the leading mechanism exhibit similar performance. This outcome is anticipated, as in this scenario, both approaches essentially align with each other and with the classical local differential privacy-based triangle counting algorithm, as described in Imola et al. (2021).

The experimental findings also reveal that, across all the download cost budgets we considered, the smooth sensitivity method consistently outperforms the clipping method.

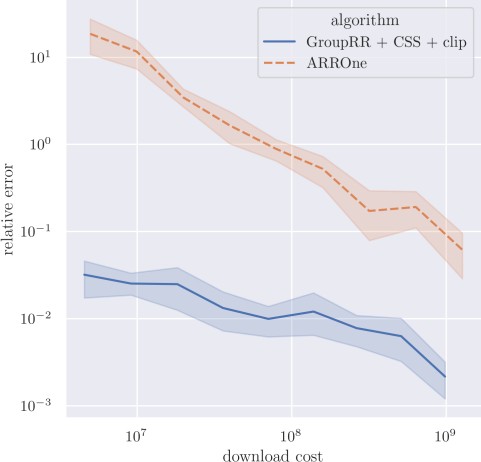

Figure 4: Comparative analysis of relative error between our algorithm and ARR across different download costs applied to the Google+ graph, assuming a privacy budget of $\varepsilon = 1$. The values of the parameter $s$ used for this experiment vary from 1 to 32, resulting in values of $\mu_c$ varying from 1 to $32,768$. The term 'CSS' in the legend stands for our central server sampling method.

Figure 4 illustrates that the performance of our method compares to the state of the art becomes better as the size of the graph increases. Indeed, as the download cost needs to be reduced by a larger factor to keep the same communication, the gap in relative errors becomes bigger. For this experience we only used the clipping version of our algorithm as smooth sensitivity is slow for large graphs.

### 7.1.2  Error Analysis for Upload Costs

Throughout this section, we fix the download-reduction target at $\mu^* = 1/1000$. As discussed in Section 7.1.1, this choice corresponds to GroupRR parameters $s = 10$ and $\mu_C = 1/10$. For ARROne, we set $\mu \approx 1/31.6$ (i.e., $\mu \approx 0.0316$).

Figure 3 demonstrates that our approach has enhanced the relative errors in counting across all ranges of upload cost budgets. Notably, the smaller the budget, the more significant the improvement we observe. For the lowest upload cost budget in this experiment, we have achieved an enhancement in relative error by up to a factor of 1000. It is also evident that the smooth sensitivity method exhibits superior performance compared to the clipping method in this context.

### 7.1.3  Error Analysis for Various Graph Sizes and Privacy Budgets

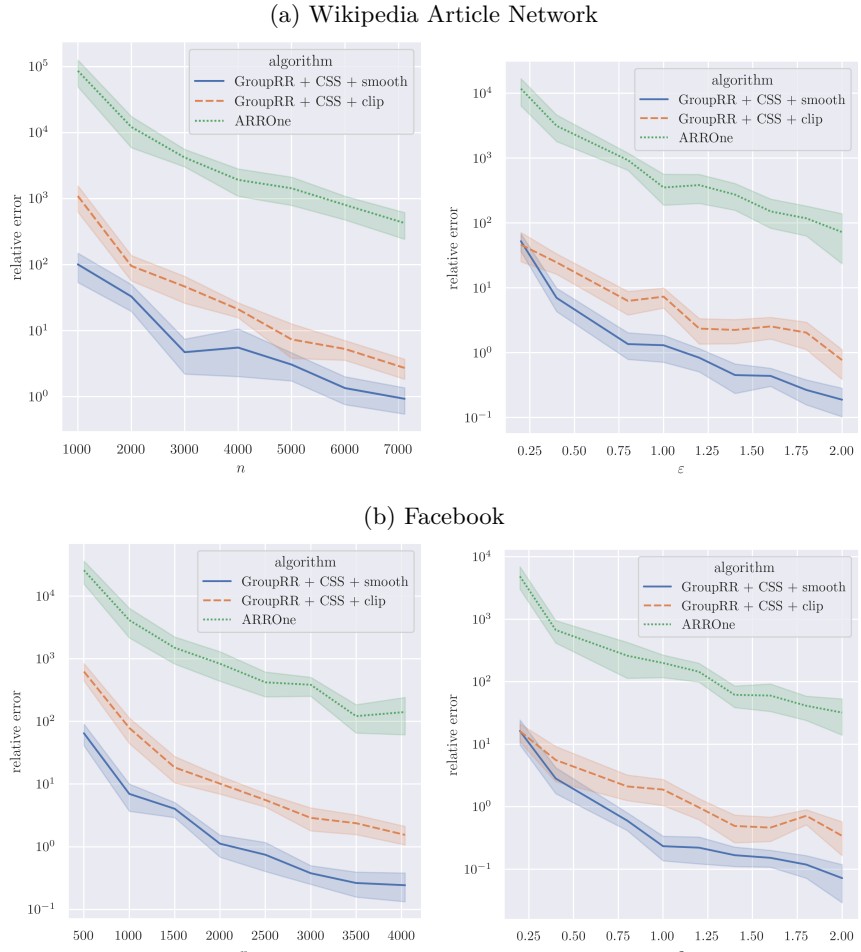

Figure 5: Comparing the relative error between our algorithm and the state-of-the-art approach (left plots) across different graph sizes ($n$), derived from the Wikipedia Article Network and Facebook graphs with $\varepsilon = 1$ (right plots) under varying privacy budgets. In both experiments, we maintain a download reduction setting of $\mu^* = 1000$. Note that in the legend, 'CSS' represents our central server sampling method.

Figure 5 shows that our algorithm improves the current state-of-the-art across a wide array of parameters. In these experiments, we tested various graph sizes and privacy budgets, observing similar trends for both algorithms. Notably, even though the performance gap between the two algorithms appears to expand as the privacy budget increases, the difference in performance consistently remains around three orders of magnitude, irrespective of the graph size.

### 7.1.4  Runtime for Various Graph Sizes

Figure 6 presents the execution times for the state-of-the-art ARR algorithm, compared to two variants of our GroupRR algorithm: one with clipping and another with smooth sensitivity, across different graph sizes. The results indicate that GroupRR with smooth sensitivity incurs significantly longer execution times, whereas GroupRR with clipping shows running times comparable to ARR. This discrepancy in performance can be attributed to their asymptotic complexities: both ARR and GroupRR with clipping operate in $\mathcal{O}(d_{max}m)$, whereas the complexity for GroupRR with smooth sensitivity is $\mathcal{O}(nm)$. The graphs, plotted on a logarithmic scale, do not form perfect straight lines, reflecting that in our model for generating graphs of varying sizes, the average degree does not scale proportionally with graph size $n$.

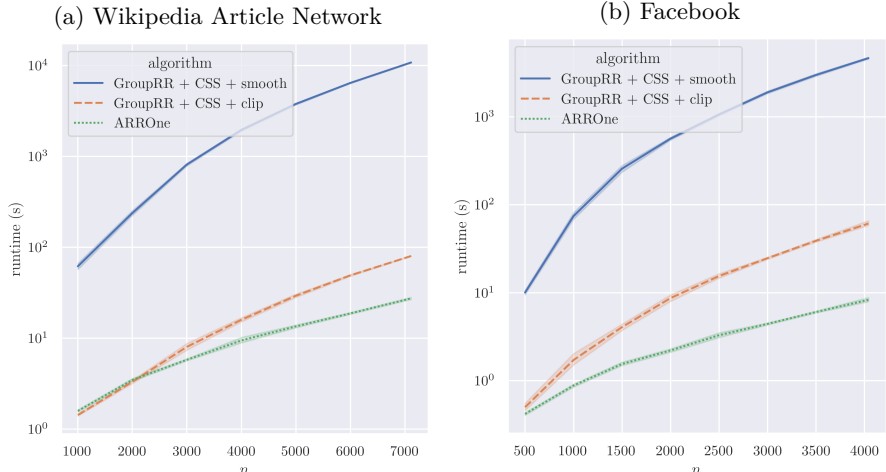

Figure 6: Comparison on runtime between our algorithm and the leading edge algorithm for various graph sizes generated from the Wikipedia Article Network and Facebook graphs with $\varepsilon = 1$ and a download cost reduction of 1000. In the legend, the abbreviation 'CSS' denotes our central server sampling method.

This experiment demonstrates the balance between speed and accuracy required when choosing between our two methods. GroupRR with clipping offers good accuracy with an execution time comparable to state-of-the-art methods. Conversely, GroupRR with smooth sensitivity prioritizes accuracy but at the expense of execution speed. This trade-off becomes particularly significant when $d_{max}$, the maximum degree, is small relative to the number of users $n$.

### 7.1.5 Error Analysis on Power-law Graphs

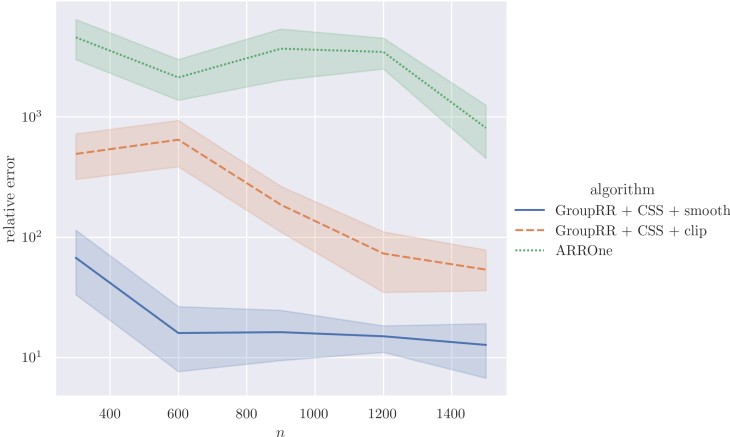

Figure 7: Comparative analysis of relative error between our algorithm and ARR across different graph sizes on synthetically generated power-law graphs, assuming a privacy budget of $\varepsilon = 1$. For this experiment, we set the parameter $s$ to 5. The term 'CSS' in the legend stands for our central server sampling method.

Figure 7 shows the relative error of our algorithms as well as the state-of-the-art ARROne for various graph sizes. Each graph is randomly samples such that the degree distribution follows a power law of exponent 2. This means that the expected number of nodes with a degree $d$ is proportional to $d^{-2}$.

Consequently, the graph's maximum degree is approximately one-third of the number of nodes. Even in this regime—where the bounded-degree condition $d_i < m \approx n/s$ required for our theoretical guarantees is violated—our algorithms still outperform the state-of-the-art ARROne.

## 7.2 4-cycles Counting

To validate the versatility of our framework, we extended our experiments to include counting cycles of length 4, or 4-cycles, utilizing the same framework employed in our triangle counting methodology. This method is described in Eden et al. (2023), Hillebrand et al. (2025) and Suppakitpaisarn et al. (2025). The number of 4-cycles that include edges $(i, i')$ and $(i, i'')$ can be estimated by $\sum_j \tilde{a}(i', j)\tilde{a}(j, i'')$ and the number of 4-cycle that the node $i$ participates in is given by $\sum_{i',i'':(i,i'),(i,i'') \in E} \sum_j \tilde{a}(i', j)\tilde{a}(j, i'')$.

Given the larger size of the subgraphs of interest, we opted to use smaller graphs for our experiments to effectively manage complexity and computational demands. To establish the robustness and applicability of our results, we conducted our 4-cycles counting experiments on two specific graphs. The first is the Twitter Interaction Network for the US Congress (Fink et al., 2023), which represents Twitter interactions of the 117th United States Congress, comprising 475 nodes and 13,289 edges. The second graph is the email-Eu-core network (Leskovec et al., 2007; Yin et al., 2017), derived from email interactions among members of a major European research institution, containing 1005 nodes and 25,571 edges. These selections allow us to provide substantive evidence of the generality of our findings across different types of networks.

We assessed accuracy using the $\ell_2$-error computed as $\sqrt{\sum_{t=1}^{10} (\hat{x}_t - x)^2}$, where $x$ is the ground truth and $\hat{x}_t$ is the estimated value obtained in trial $1 \le t \le 10$.

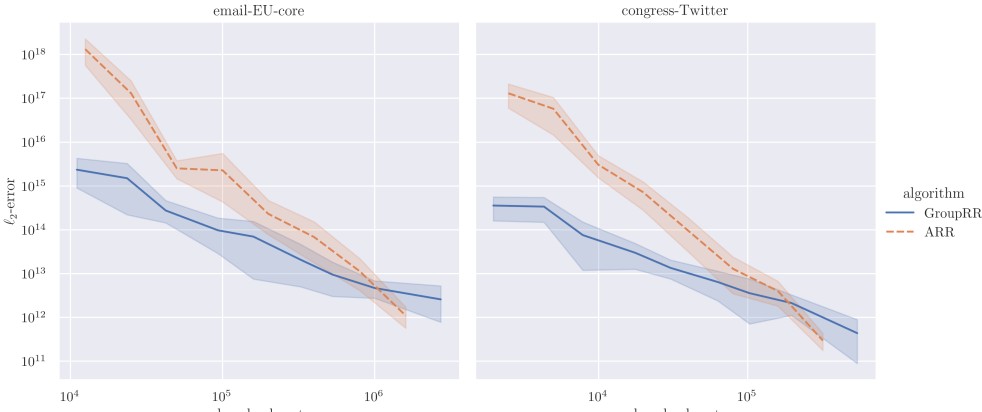

Figure 8: Comparison on $\ell_2$-error between our algorithm and ARR for various sampling factors on the Twitter Interaction Network for the US Congress and the email-Eu-core network with $\varepsilon = 1$.

The experimental outcomes are displayed in Figure 8. It is evident that the error patterns are consistent across both datasets. With either method, regardless of the sampling factor used, the error is greater in the email-EU-core dataset. This aligns with expectations since the $\ell_2$-error is a property that tends to increase with the graph's size.

Nonetheless, the rate at which the discrepancy between the two methods grows is the same across both datasets. When the full dataset is used—i.e., without any communication reduction—the performance of ARR and GroupRR is comparable and aligns with that of the standard algorithm without communication constraints (Imola et al., 2021). However, as shown in the graphs, under a sampling factor of 100, the error associated with ARR increases by approximately a factor of $10^3$ relative to GroupRR. According to Theorem 4, the $\ell_2$-error in estimating a single edge scales linearly with the sampling factor $s$ for GroupRR, but quadratically (i.e., as $s^2$) for ARR. Since each 4-cycle estimation involves two edge estimates, the overall $\ell_2$-error scales as $s^2$ for GroupRR and as $s^4$ for ARR. This implies a theoretical error gap of $s^2 = 10^4$ between

the two methods. The observed reduction of this gap to $10^3$ is attributed to the fact that, in the absence of sampling, ARR achieves slightly better accuracy than GroupRR.

## 8    Conclusion

Our work introduces a private graph publishing mechanism that provides unbiased estimations of all graph edges while maintaining low communication costs. The effectiveness of this method stems from integrating two key components: the application of linear congruence hashing to achieve uniform edge partitions and the amplification of budget efficiency through sampling within each group. This synergy leads to a significant reduction in download costs, quantifiable as a factor of $\mathcal{O}(s^3)$, where $s$ represents the size of the groups.

We then demonstrated its utility in accurately estimating the count of triangles and 4-cycles. Furthermore, we elaborated on the application of smooth sensitivity to these problems, ensuring that the resulting estimations remain unbiased. Our experiments have demonstrated that these methods offer substantial improvements in precision over existing state-of-the-art approaches for the countings.

Our future work is focused on enhancing the scalability of subgraph counting under local differential privacy. We have observed that the computation time required for each user in all mechanisms proposed to date is considerable. This computation time often increases for mechanisms involving multiple steps. Our aim is to explore how sampling methods can not only improve precision but also significantly boost the scalability of the counting process.

## 9    Acknowledgments

Quentin Hillebrand is partially supported by KAKENHI Grant 20H05965, and by JST SPRING Grant Number JPMJSP2108. Vorapong Suppakitpaisarn is partially supported by KAKENHI Grant 21H05845 and 23H04377. Tetsuo Shibuya is partially supported by KAKENHI Grant 20H05967, 21H05052, and 23H03345. The authors wish to express their thanks to the anonymous reviewers whose valuable feedback greatly enhanced the quality of this paper.

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
