# OpenReview forum: "Communication Cost Reduction for Subgraph Counting under Local Differential Privacy via Hash Functions"
_TMLR — Accepted by TMLR_

### Review · Reviewer_1AKE · 2025-06-01

**Summary Of Contributions:**

This paper considers important problems in subgraph counting (including the number of triangles which is a key graph statistics, the clustering coefficient) under differential privacy.  The model is each user knows its edges, we are asked to compute global statistics.  Under Local Differential Privacy, the users do not share their data in the raw (typically add Laplacian noise to it) to even ensure privacy from the global entity computing the stats.

This paper provides a new protocol for these tasks.  It is multiple round, involving the users (vertices) sending noising information to central node, the central node adding additional global noise, and sending tasks back to users to compute statistics, and sending them back to be compiled globally.  The main result uses linear congruence hashing to deliver sampled estimates in a way that under certain settings provides much better bounds on the estimates in theory, and in practice.

**Audience:**

Yes

**Claims And Evidence:**

Yes

**Requested Changes:**

I have on additional question:

Questions:
  - Some of the communication consequences are directly from the degree < m < n/s assumption.  Can these be replaced by average case bounds if it is true in average?  For instance this seems to be the case in Corollary 1.  Can this idea be used generally to remove this strong degree assumption?


Regardless of the above weaknesses (and question), I think the paper is valuable.  With a few small changes I think it's worth publishing.  It's on an interesting relevant topic, and it is generally very well written.  To get my final support for the paper, please respond (and update the paper as/if needed) the weaknesses and Question.

**Strengths And Weaknesses:**

Strengths:
 - the paper is very well written, with formal definitions and details that made most of the statements clear and easy to follow
 - it attacks and important problem, and provides (in reasonable cases) clear advantages to state-of-the-art
 - the advantages are not just theoretical (the main results), but also sees to work much better in practice on a couple of examples.

Weakness:
 - the main results seem to leverage on the assumption that all nodes have degree bounded by n/s where s is the size of the sample.  This is stated as reasonable without describing or citing justification.  My understanding is that while this may be true on average, or in some cases, it is not in other cases as well.  (e.g., Elon apparently has over 200 million followers on Twitter/X).  While this assumption is stated in main theorems, I think it should be discussed more clearly in the intro, and authors should add more discussion on when it holds, and what could be done if it does not hold.
  - A few proofs could have explained steps a bit better.  Namely in proof of Theorem 4, it would have been useful, immediately below the last offset equation, to remind the reader than Lemma 5 and the m < n/s assumption are being used.

---

> ### Author Response · Authors · 2025-06-18
>
> Thank you very much for your kind consideration and suggestions on our paper.
>
> > the main results seem to leverage on the assumption that all nodes have degree bounded by n/s where s is the size of the sample. This is stated as reasonable without describing or citing justification.
>
> We have the following discussion before Lemma 5.
>
> > In Lemma 2 and later, we work under the assumption that $d_i < m \approx n/s$. This assumption is reasonable for almost all social networks, where $d_i$ typically remains constant, making this assumption applicable in most cases.
>
> We appreciate your thoughtful suggestion and agree that the previous justification requires appropriate citations. We will include them in the next version of the manuscript.
>
> > My understanding is that while this may be true on average, or in some cases, it is not in other cases as well. (e.g., Elon apparently has over 200 million followers on Twitter/X).
>
> The social network considered in this work is undirected. As the reviewer rightly noted, in a directed network, some nodes may indeed have a large indegree. However, we believe this concern is less relevant in the undirected case, where relationships are inherently reciprocal. We will add this discussion in the next version of this manuscript.
>
> > While this assumption is stated in main theorems, I think it should be discussed more clearly in the intro.
>
> Thank you very much for your kind suggestions regarding to our assumption on $d_i$. We agree with the reviewer that this assumption should be stated in the introduction of the paper.
>
> > Authors should add more discussion on when it holds, and what could be done if it does not hold.
>
> We emphasize that the privacy guarantees of our protocol, as discussed in Section 5.1, do not rely on the assumption $d_i < m$; relaxing this condition does not compromise privacy.
>
> Also, the assumption $d < m$ is not required for the communication complexity result stated in Theorem 6 of Section 5.5. As shown in the theorem, our improvement in communication complexity holds with an $s$-fold reduction, when $d_i$ is larger than $n / s$.
>
> The assumption $d < m \approx n/s$ is invoked in Section 5.3 solely to support our theoretical analysis of accuracy. Specifically, it is used to derive an upper bound on $\sigma_i$ in Lemma 5 (Page 12). Since $\sigma_i$ plays a central role in analyzing the variance and covariance of the publication mechanism, omitting this assumption makes it difficult to theoretically claim that the variance increases by only a factor of $s$. To address the absence of a theoretical upper bound in this case, we plan to include an experimental evaluation of the variance and accuracy under this relaxed condition in the next version of our manuscript.
>
> > A few proofs could have explained steps a bit better. Namely in proof of Theorem 4, it would have been useful, immediately below the last offset equation, to remind the reader than Lemma 5 and the m < n/s assumption are being used.
>
> Thank you very much for your kind suggestion. We will revise our proofs to ensure that all steps are clearly explained.
>
> > Some of the communication consequences are directly from the degree < m < n/s assumption. Can these be replaced by average case bounds if it is true in average? For instance this seems to be the case in Corollary 1. Can this idea be used generally to remove this strong degree assumption?
> Unfortunately, the degree assumption cannot be replaced by average-case bounds. This is because a small number of nodes with high degrees can lead to large upper bounds on the covariance, which may dominate the contributions from other nodes.

---

> ### Author Response · Authors · 2025-07-04
>
> With the end of the discussion period approaching, I wanted to ensure that my rebuttal was considered as part of the review process. Could you please let me know there remain any non-addressed concerns about the article?
>
> Thank you for your time

---

> > ### Comment · Reviewer_1AKE · 2025-07-04
> >
> > Thanks for the reminder.  You have made suggestions of how to change the paper, but they are vague.  Can you be more precise?  For instance:
> >   - what paper will you cite to support the claim that the maximum degree is often bounded?  Is there such a reference that would demonstrate this (the lack of specifics in your response has increased my doubt about this assumption)
> >   - why do undirected networks make the degree problem better?  I can turn a directed graph into an undirected one by making an edge if one exists in either direction.  I do not understand what you have in mind here.
> >   - did you make any progress on the experimental evaluation you planned to include?  Can you show me something to convince me that the experiments will support your claim? I am not confident they will, since it does not align with the theory.
> >  - how will you revise the proofs to make them more clear?   Can you convince me here in the comments that you will be able to accomplish this?

---

> > > ### Author Response · Authors · 2025-07-07
> > >
> > > Thank you very much for your reply and suggestions on our paper.
> > >
> > > > why do undirected networks make the degree problem better? I can turn a directed graph into an undirected one by making an edge if one exists in either direction. I do not understand what you have in mind here.
> > >
> > > Although, as the reviewer noted, any directed graph can be transformed into an undirected one, our focus is on graphs that are naturally undirected. In such graphs, we believe the maximum degree is inherently limited by physical or practical constraints. For instance, if we consider a graph representing physical interactions over the past two weeks--relevant in contexts such as epidemic modeling--the number of interactions any individual can have is bounded and cannot approach the size of the global population.
> > >
> > > Platforms like X, which represent relationships using directed graphs, can exhibit extremely high out-degrees (e.g., Elon Musk’s follower count). However, in social networks where relationships must be mutual, such as Facebook, the degree of each node tends to remain modest. For example, Facebook imposes a hard cap of 5,000 friends per user [a].
> > >
> > > Additionally, using GRR + CSS, the actual communication reduction is in orders of $s^3$ when the degree is bounded on average. Thus, even in cases where the maximal degree represents a significant fraction of the number of nodes, a meaningful reduction of the communication cost can still be achieved.
> > >
> > >
> > > > did you make any progress on the experimental evaluation you planned to include? Can you show me something to convince me that the experiments will support your claim? I am not confident they will, since it does not align with the theory.
> > >
> > > We have conducted additional small-scale experiments on synthetic graphs designed to satisfy a power-law distribution of degrees.
> > > We generate power-law graphs for different sizes with:
> > >
> > > ```
> > > import networkx as nx
> > > s = nx.utils.powerlaw_sequence(size, 2)
> > > g = nx.expected_degree_graph(s, selfloops=False)
> > > ```
> > >
> > > The obtained graphs had an average maximum degree of around a third of the number of nodes.
> > > The relative errors (each cell is obtained by averaging across 10 runs) for 3 different algorithms each run with a communication cost reduction of 100 are shown in the following tables.
> > >
> > > |                | 300  | 600  | 900  | 1200 | 1500 |
> > > |----------------|------|------|------|------|------|
> > > | grr_css_smooth | 67   | 16   | 13   | 15   | 13   |
> > > | grr_css_clip   | 493  | 647  | 186  | 73   | 54   |
> > > | arr_one        | 4606 | 2135 | 3707 | 3481 | 812  |
> > >
> > > This table shows that even in the case were the assumptions that $d_{max} < n / s$ doesn’t hold, our algorithm still performs significantly better than the state-of-the-art communication efficient triangle counting methods.
> > >
> > >
> > > > how will you revise the proofs to make them more clear? Can you convince me here in the comments that you will be able to accomplish this?
> > >
> > > The requested clarification is to explicitly state the assumption immediately before it is used.
> > >
> > > > Namely in proof of Theorem 4, it would have been useful, immediately below the last offset equation, to remind the reader than Lemma 5 and the m < n/s assumption are being used.
> > >
> > > Consequently, we will include this assumption explicitly in the statement of Theorem 4.
> > >
> > >
> > > [a] What to Do When Your Facebook Profile is Maxed Out on Friends, https://authoritypublishing.com/social-media/what-to-do-when-your-facebook-profile-is-maxed-out-on-friends/

---

> > > > ### Comment · Reviewer_1AKE · 2025-07-08
> > > >
> > > > Thank you for the careful response.  And please excuse my delayed reply to your response to my review.

---

### Review · Reviewer_mqCD · 2025-06-02

**Summary Of Contributions:**

In this submission, the authors propose a way to perform subgraph counting subject to edge local differential privacy. Prior approaches relied upon a naive use of randomized response: each bit in a client's adjacency list is flipped with probability $1/\exp(\varepsilon)$. This has the effect of being both noisy and costly in terms of communication.

The submission utilizes subsampling to improve both. Specifically, the central server sends a linear congruence hash function to each client. The client uses the hash to essentially color the indices of their adjacency list then, for every color, they choose a random index of that color and apply randomized response to the value at that index. The fact that an adversary cannot see which entry was sampled is used to minimize the noise in the entry itself.

To count triangles in their neighborhood, a client re-scales and shifts the bits received from other clients. They communicate this count to the server via the Laplace or smooth sensitivity mechanism. Experiments reveal that the new work dominates the prior work's tradeoff between error and epsilon and the tradeoff between error and download cost. Running time however is worsened.

**Audience:**

Yes

**Claims And Evidence:**

Yes

**Requested Changes:**

See above.

Also,
- Make it a bit clearer early on that you will focus on triangle counting instead of leaving the goal as the generic-sounding "subgraph counting."
- Consider renaming "the clipping method" to "the global sensitivity method", to better parallel "smooth sensitivity" (or just Laplace mechanism!)
- Please clarify the difference between "ARR" and "ARROne."

**Strengths And Weaknesses:**

The task of subgraph counting is a fundamental one and the setting of edge local DP is likewise textbook. The focus on communication cost is proportionate to its importance. The pseudocode and Section 4 in general do a good job of conveying to the reader (a) the main ideas of the proposed local protocol and (b) basic intuition about why the proposal might work.

I would have liked more advanced intuition. Client i makes a guess about whether a triangle i,j,k is present (or subgraph more generally) based on a noisy version of the indicator of (j,k)'s presence *that is diluted* via subsampling; how does the scaling and shifting of noisy sums compensate for the fact that client i could get a noisy version of the wrong bit?

A key hyperparameter in the new algorithm is s, the target size of each group that the hash function divides nodes into. However, the experimental section does not spell out the exact values of s used. (I know that Section 7.1.1 contains a formula but it superficially appears circular---$\mu^\*$ is defined in terms of $s^2$ but then $s$ is set in terms of $\mu^\*$--- which confuses the reader.)

---

> ### Author Response · Authors · 2025-06-18
>
> Thank you very much for your kind consideration and suggestions on our paper.
>
> 1. Advanced intuition
>
> > Client i makes a guess about whether a triangle i,j,k is present (or subgraph more generally) based on a noisy version of the indicator of (j,k)'s presence that is diluted via subsampling; how does the scaling and shifting of noisy sums compensate for the fact that client i could get a noisy version of the wrong bit?
>
> Thank you very much for your kind comment. We agree that this intuition is missing in the current version of our manuscript, and we will include it in the next revision. As you rightly pointed out, client i may receive an incorrect bit with some probability. However, due to the amplification effect from subsampling, the bit obtained becomes less noisy under the same privacy budget. Moreover, this probability was computed in the article and the unbiasing takes it into account. As a result, the estimator provided for the estimation of the presence of an edge is fully unbiased. This leads to a more accurate estimate giving the communication complexity or the privacy budget.
>
>
> 2. Hyperparameter $s$
>
> > A key hyperparameter in the new algorithm is s, the target size of each group that the hash function divides nodes into. However, the experimental section does not spell out the exact values of s used.
>
> We regret that the value of $s$ was not explicitly stated in Section 7. In Figures 3 and 4, we vary the value of $s$ used in the experiments, with each point in the graphs corresponding to a specific value of $s$. We will specify the value of $s$ in these two figures in the next version of this manuscript.
>
> > (I know that Section 7.1.1 contains a formula but it superficially appears circular---$\mu^*$ is defined in terms of $s^2$ but then $s$ is set in terms of $\mu^*$--- which confuses the reader.)
>
> Thank you very much for your kind comment. We have the following statement in Section 7.1.1.
>
> > Consider the reduction in download costs, denoted by $\mu^*$, which equals to $\mu_c / s^2$ for GroupRR with central server sampling. Setting $s = (1 / \mu^)^{1/3}$ and $\mu_c = (\mu^*)^{1/3}$, we observe that the $S_2$ factor in the error, as discussed in Theorem 11, grows with $(\mu^*)^{-2/3}$, while the $C_4$ factor grows with $(\mu)^{-1/3}$.
>
> In other words, our analysis in Section 6 shows that the communication cost $\mu^*$ is approximately given by $\mu_c / s^2$. In the experiment presented in Figure 4, we fix $\mu^* = 1000$ and then select values of $\mu_c$ and $s$ such that $\mu_c / s^2 = 1000$. Specifically, we chose $\mu_c = 10$ and $s = 1/10$ for this setting. We acknowledge that our explanation in the current manuscript is not sufficiently clear, and we will revise it to improve clarity in the next version.
>
>
> 3. Difference between “ARR” and “ARROne”
>
> “ARROne” is the variant of “ARR” that uses the 4-cycle trick described in Section 3. We will make it clear in the section that we use this denomination for the rest of the article.
>
> Other comments
>
> > Make it a bit clearer early on that you will focus on triangle counting instead of leaving the goal as the generic-sounding "subgraph counting."
>
> > Consider renaming "the clipping method" to "the global sensitivity method", to better parallel "smooth sensitivity" (or just Laplace mechanism!)
>
> We agree with both of the comments, and will make the suggested modifications in the next version.

---

> > ### Author Response · Authors · 2025-06-26
> >
> > We noticed that the second section of our reply wasn't rendered correctly so we repost it for easier reading.
> >
> > 2. Hyperparameter $s$
> >
> > > A key hyperparameter in the new algorithm is s, the target size of each group that the hash function divides nodes into. However, the experimental section does not spell out the exact values of s used.
> >
> > We regret that the value of $s$ was not explicitly stated in Section 7. In Figures 3 and 4, we vary the value of $s$ used in the experiments, with each point in the graphs corresponding to a specific value of $s$. We will specify the value of $s$ in these two figures in the next version of this manuscript.
> >
> > > (I know that Section 7.1.1 contains a formula but it superficially appears circular---$\mu^{\star}$ is defined in terms of $s^2$ but then $s$ is set in terms of $\mu^{\star}$--- which confuses the reader.)
> >
> > Thank you very much for your kind comment. We have the following statement in Section 7.1.1.
> >
> > > Consider the reduction in download costs, denoted by $\mu^{\star}$, which equals to $\mu_c / s^2$ for GroupRR with central server sampling. Setting $s = (1 / \mu^{\star})^{1/3}$ and $\mu_c = (\mu^{\star})^{1/3}$, we observe that the $S_2$ factor in the error, as discussed in Theorem 11, grows with $(\mu^{\star})^{-2/3}$, while the $C_4$ factor grows with $(\mu)^{-1/3}$.
> >
> > In other words, our analysis in Section 6 shows that the communication cost $\mu^{\star}$ is approximately given by $\mu_c / s^2$. In the experiment presented in Figure 4, we fix $\mu^{\star} = 1000$ and then select values of $\mu_c$ and $s$ such that $\mu_c / s^2 = 1000$. Specifically, we chose $\mu_c = 10$ and $s = 1/10$ for this setting. We acknowledge that our explanation in the current manuscript is not sufficiently clear, and we will revise it to improve clarity in the next version.

---

> ### Author Response · Authors · 2025-07-04
>
> With the end of the discussion period approaching, I wanted to ensure that my rebuttal was considered as part of the review process. Could you please let me know there remain any non-addressed concerns about the article?
>
> Thank you for your time

---

### Review · Reviewer_e61A · 2025-06-21

**Summary Of Contributions:**

The authors propose a novel approach to efficiently compute subgraph statistics, such as triangle and 4-cycle counts, under edge local differential privacy by introducing a technique called Group Randomized Response (GroupRR). This mechanism leverages linear congruence hashing to partition users into reproducible groups and sample one representative edge per group, which is then privatized using a randomized response mechanism. This sampling strategy enables privacy amplification while significantly reducing communication overhead by a factor of s^2, with only a linear increase in variance. The method is shown to be unbiased, scalable, and generalizable to various subgraph counting tasks, outperforming previous state-of-the-art algorithms by up to three orders of magnitude in terms of ℓ2-error when communication cost is held constant.

**Audience:**

Yes

**Claims And Evidence:**

Yes

**Requested Changes:**

1. Increased Estimation Variance. I can easily see that there is a trade-off between communication cost and variance: while communication cost is reduced by a factor of $s^2$, variance increases. So, a critical question is how to select the best s? I think the authors need to give some theoretical results and discussions.

2. Parameter Sensitivity. The performance depends heavily on careful tuning of the sampling size s, privacy budget split $(\epsilon_0, \epsilon_1, \epsilon_2)$, and hashing parameters but it does not offer clear guidance or adaptive strategies for selecting these parameters in practice.

3. While smooth sensitivity offers improved utility, its computation can be expensive, especially when $\gamma$ must be large to control variance.

**Strengths And Weaknesses:**

1. The core contribution is a mechanism that reduces communication costs by a factor of $s^2$ using linear congruence hashing and sub-sampling, which is a major bottleneck in prior ELDP-based subgraph counting methods.

2. The method achieves amplification of privacy guarantees due to subsampling, which allows for lower bit-flipping probabilities in the randomized response mechanism.

3. The proposed estimator is shown to be unbiased, meaning it correctly estimates subgraph counts in expectation despite local randomization and compression.

4. Empirical results show that for fixed communication cost, the proposed method achieves up to 1000x lower ℓ2-error in triangle counting compared to state-of-the-art approaches.

---

> ### Author Response · Authors · 2025-06-26
>
> Thank you very much for your kind consideration and suggestions in our paper. Please kindly find the answers to your questions below:
>
> > Increased Estimation Variance. I can easily see that there is a trade-off between communication cost and variance: while communication cost is reduced by a factor of $s^2$, variance increases. So, a critical question is how to select the best $s$? I think the authors need to give some theoretical results and discussions.
>
> We believe that the value of $s$ should be determined based on the user's communication budget. Since the communication cost is on the order of $n^2$, where $n$ is the number of nodes, our algorithm reduces the communication cost to $n^2 / s^2$. Given a communication budget of $M$, it is therefore appropriate to set $s = n / \sqrt{M}$. Our experimental results demonstrate that, under the same communication budget, our approach achieves lower variance compared to prior work. Although we have addressed the selection of $s$ in Sections 6 and 7.1.1, we recognize from this comment that our explanation may be insufficient. We will expand on this discussion in the next version of the paper.
>
> > The performance depends heavily on careful tuning of the sampling size s, privacy budget split $(\varepsilon_0, \varepsilon_1, \varepsilon_2)$, and hashing parameters but it does not offer clear guidance or adaptive strategies for selecting these parameters in practice.
>
> We provide an error bound as a function of $s$, $\mu_c$, $\varepsilon_1$, $\varepsilon_2$, and graph statistics in Theorems 12 and 13. If the operator has access to certain information about the graph, they can choose the parameters to minimize this bound. In general, a small value of $\varepsilon_0$ does not significantly impact the result, whereas small values of $\varepsilon_1$ and $\varepsilon_2$ do. Therefore, when the relevant graph information required by Theorems 12 and 13 is unavailable, we recommend allocating a small portion of the privacy budget to the degree publishing step ($\varepsilon_0$) and dividing the remaining budget equally between the two main steps of the algorithm ($\varepsilon_1$ and $\varepsilon_2$). We will include this discussion in the next version of this manuscript.
>
> > While smooth sensitivity offers improved utility, its computation can be expensive, especially when $\gamma$ must be large to control variance.
>
> As mentioned at the end of Page 5, we use $\gamma = 4$, which is the smallest value of $\gamma$ for which we can effectively control the variance. For small graphs, the execution time of the smooth sensitivity mechanism was manageable with $\gamma = 4$ (see Figures 3 and 5). For larger graphs, we propose a faster alternative that uses clipping. While this approach results in a slight reduction in accuracy, it significantly improves computational efficiency (see Figure 4).

---

> ### Author Response · Authors · 2025-07-04
>
> With the end of the discussion period approaching, I wanted to ensure that my rebuttal was considered as part of the review process. Could you please let me know there remain any non-addressed concerns about the article?
>
> Thank you for your time

---

### Decision · Action_Editor_vBXu · 2025-08-05

**Recommendation:** Accept as is

**Audience:**

Yes

**Audience Explanation:**

The subset of the community that cares about privacy and security of machine learning would find the paper interesting.

**Claims And Evidence:**

Yes

**Claims Explanation:**

The reviewers appear to be unanimous that all claims in the paper are well supported by evidence. I agree, and I think the paper should be accepted.